# DMIL-Net: A Multi-View Fusion and Region Decoupling Network For Diffusion-Based Generative Image Forgery Localization

## Abstract

The iteration and popularization of diffusion models have significantly lowered the barrier to high-quality image forgery, posing severe challenges to image authenticity forensics. Addressing local forgery based on diffusion models, this paper proposes a forgery localization method named DMIL-Net. Specifically, we first design a multi-view feature learning strategy that integrates RGB views, noise views, and high-frequency views, aiming to capture diffusion model-specific edge fusion artifacts and denoising artifacts generated during the generation process, thereby providing clues for accurate localization. Secondly, considering the inherent differences in artifact features between the main content regions and edge detail regions generated by diffusion models, we propose a tampered region decoupling and integration strategy. This strategy iteratively decouples and integrates the main regions and detail regions to achieve more precise localization. In addition, we construct the DMI dataset, which contains 50,000 generative forgery images created via five prevalent diffusion-based generative image forgery methods, to support model training and testing. Experimental results show that DMIL-Net outperforms five mainstream methods on localization performance, generalization, extensibility, and robustness.

## 1 Introduction

Artificial intelligence is revolutionizing the field of visual creation with the advent of diffusion model-based techniques like Stable Diffusion Rombach et al. (2022). These methods simulate physical diffusion to learn complex data structures and create new samples, showing significant potential in image synthesis by generating diverse styles and detailed artworks. Local image generation, which fills in missing or damaged image areas, is a notable application that uses surrounding pixel information to produce coherent and realistic content. This technology excels in quality and style preservation, benefiting art restoration and digital editing. However, it also presents challenges in image authenticity verification due to its potential misuse for tampering, making it hard even for experts to differentiate between authentic and AI-generated content, thus posing new challenges to forensics.

Mainstream image tampering localization (ITL) methods are inadequate for addressing the task of diffusion-based generative image forgery localization (DMIL). Diffusion models preserve image details and structures so well that tampered regions closely resemble originals, challenging mainstream ITL methods that rely on traditional noise view strategies Dong et al. (2022) and frequency domain strategies Wang et al. (2022a) for accurately detecting tampering traces. Moreover, the seamless integration of tampered regions with the background in diffusion model outputs lacks clear boundaries, further complicating the localization process. These diffusion-generated images often lack the distinct edges that mainstream ITL methods depend on, resulting in localization outcomes lacking in edge detail. Lastly, lacking a large-scale dataset for DMIL hampers research and model training in this field. Creating a high-quality DMIL dataset requires extensive annotation efforts and must simulate the characteristics of artificially tampered images to ensure its effectiveness and practicality.

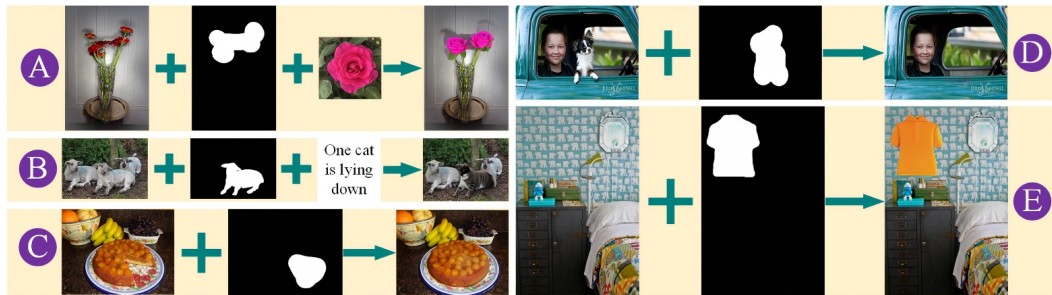

Figure 1: Five tampering types in DMI dataset. A: Example-guided, B: Text-guided, C: Context-aware, D: Object removal, E:Shape-guided.

In response to existing challenges, we introduce DMIL-Net, whose structure is shown in Figure 2. Specifically, we divide the DMIL task into two stages: Capturing Tamper Traces and Localizing Tampered Regions. In the first stage, we first propose a multi-view feature learning strategy. Targeting the characteristics of diffusion models, we innovatively introduce a novel noise view extractor and a high-frequency information extractor. We comprehensively capture clues from tampered regions by combining RGB views, noise views, and high-frequency information features. Secondly, we propose a multi-level contrastive learning strategy. On the one hand, it better captures the long-term dependencies between the three modalities, generating more comprehensive fused features. On the other hand, it calculates pixel-level contrastive losses of multi-view fusion features at multiple levels, forming a deep understanding of the scene. In the second stage, we decouple the tampered region into a body region and a detail region, with the body region concentrated in the center of the tampered region and the detail region composed of pixels around the edge. We design three decoder branches, two of which analyze the body and detail region of the tampered region, and the third branch integrates the output features of the two decoupled branches to generate complete and detail-rich localization results. To support model training, we constructed a large-scale dataset called DMI. It covers five mainstream diffusion-based generative local image forgery methods, and includes five common forms of tampering as depicted in Figure 1, offering comprehensiveness and diversity. It contains 50,000 tampered images with diverse scenarios, and most masks are manually drawn to simulate real-world tampering.

The main contributions of this work include: (1) We propose DMIL-Net, a novel generative image forgery localization model that integrates multi-view feature learning, and a tampering region decoupling-integration strategy. (2) We construct a high-quality DMI dataset, characterized by its substantial scale, diversity, and comprehensiveness. (3) Extensive experiments demonstrating the superior performance of DMIL-Net over five mainstream methods in localization precision, generalization, extensibility, and robustness.

## 2 RELATED WORK

We categorize mainstream ITL methods into four main groups: traditional methods Ren et al. (2023); Liu et al. (2022), noise-assisted methods Dong et al. (2022); Niloy et al. (2023); Lin et al. (2023); Wu et al. (2019); Xu et al. (2023); Ji et al. (2023); Wang et al. (2023), frequency domain-assisted methods Wang et al. (2022a); Xu et al. (2023); Liu et al. (2023); Kwon et al. (2021), and edge-assisted methods Dong et al. (2022); Lin et al. (2023); Zhang et al. (2021); Shi et al. (2023). Traditional methods primarily rely on attribute differences between tampered and untampered regions in RGB image features. By learning these attribute differences, models can detect tampered regions. However, image forgeries generated by diffusion models are almost indistinguishable. Noise-assisted methods aim to improve detection accuracy by combining RGB space features with noise views. Noise views generated through operations like BayarConv Dong et al. (2022) or SRM filter Niloy et al. (2023) can leverage differences in noise distribution between new elements introduced by forgery operations and authentic regions Dong et al. (2022); Wu et al. (2019); Zhou et al. (2018). Frequency domain-assisted methods mainly use DCT Wang et al. (2022a); Xu et al. (2023) or FFT Liu et al. (2023) to extract features, capturing subtle forgery traces not visible in the RGB domain Kwon et al. (2021); Arpita et al. (2019). However, most existing models do not fully capture the long-term de-

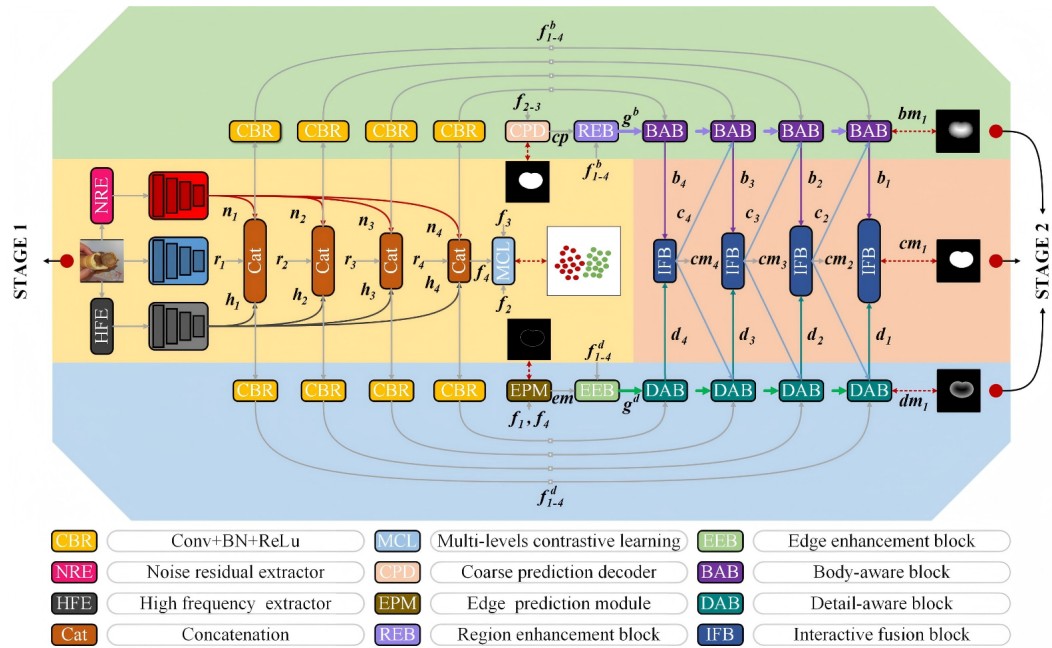

Figure 2: The structure of DMIL-Net, components include: Noise Residual Extractor (NRE), High Frequency Extractor (HFE), Multi-Level Contrastive Learning (MCL), Coarse Prediction Decoder (CPD), Edge Prediction Module (EPM), Region Enhancement Block (REB), Edge Enhancement Block (EEB), Body-Aware Block (BAB), Detail-Aware Block (DAB), and Interaction Fusion Block (IFB).

pendencies between multi-view features, failing to achieve effective interaction and fusion between multi-view features. Moreover, the design of ineffective noise/frequency domain view generators often introduces redundant information, causing adverse interference in localization results. Image tampering can lead to edge inconsistencies, and edge assistance can capture these inconsistencies, thereby improving the accuracy of tampering localization Lin et al. (2023). However, forgery images generated by diffusion models does not have obvious boundaries. A single and overconfident edge prior can often mislead the model into generating incorrect localization results.

## 3 METHOD

### 3.1 CAPTURING TAMPER TRACES

Tampering regions generated by diffusion models may have noise patterns that differ from those of authentic regions, which can serve as clues of tampering. Inspired by Frick & Steinebach (2024), we designed a Noise Residual Extractor (NRE). Specifically, as shown in Figure 3, first, the NRE uses a non-local means filter as a denoising method to generate a noise-free clean view corresponding to the original tampered image. Then, the original tampered image is subtracted from the clean view to obtain the noise residual. Finally, to further enhance the difference areas, we multiply the values of the noise residual map by 100. This process can be described as:

$$NV = (I - \text{Denoise}(I)) \otimes 100. \tag{1}$$

where, $NV$ refers to the noise view, $I$ refers to the original tampered image, $\text{Denoise}(\cdot)$ refers to the non-local means filter, $\otimes$ denotes the multiplication operation.

High-frequency features contain details such as edges, textures, and patterns, which are crucial for identifying subtle changes within the image. In image tampering forensics, the loss or anomaly of these details is often direct evidence of tampering Guo et al. (2023). We designed a High-Frequency Extractor (HFE) that applies the Laplacian of Gaussian (LoG) function to the original RGB tampered



Figure 3: Noise view extraction process.

image to obtain high-frequency information $HF$. This process can be described as:

$$HF = I - U_{\times 2}(D_{\times 2}(G(I))). \tag{2}$$

where, $G(\cdot)$ refers to the convolution operator with a Gaussian filter, $U(\cdot)$ denotes the upsampling operation, and $D(\cdot)$ denotes the downsampling operation.

We adopt three PVTv2s Wang et al. (2022b) as the backbones of DMIL-Net, feeding the original RGB tampered image $I$, high-frequency information $HF$, and noise view $NV$ into respective backbone networks to extract tampering features $r_i, n_i, n_i, i = \{1, 2, 3, 4\}$. Features from the same level across the three modalities are then concatenated, and CBRs (Convolution, Batch Normalization, and ReLU) are applied to adjust the channel number of the fused features to 64.

$$f_i = C_{1 \times 1}\big(\text{Cat}(h_i, r_i, n_i)\big), \quad i = 1, 2, 3, 4. \tag{6}$$

where, $C_{1 \times 1}(\cdot)$ refers to $1 \times 1$ CBR, $\text{Cat}(\cdot)$ refers to the concatenation operation. Contrastive learning leverages the comparison of positive and negative sample pairs to learn effective representations of data. In the feature fusion task of three views, it can better capture the long-term dependencies between the three modalities and generate more comprehensive fused features. Inspired by Niloy et al. (2023); Xiao et al. (2024); Wang et al. (2021), we propose a Multi-Level Contrastive Learning (MCL) strategy. First, we apply the contrastive learning strategy to fused features from multiple levels, which helps the model to form an in-depth understanding of the scene at each level, thereby more accurately localizing the tampered regions. As shown on the left of Figure 4, we select $f_{2-4}$ as the feature inputs for the MCL module, and use different numbers of $4 \times 4$ transpose convolutions with two strides to upsample $f_2$, $f_3$, and $f_4$ by 2, 4, and 8 times, respectively.

$$\begin{cases} f_2^{\uparrow 2} = C_{1 \times 1}(TC_{4 \times 4}(f_2)), \\ f_3^{\uparrow 4} = C_{1 \times 1}(TC_{4 \times 4}(TC_{4 \times 4}(f_3))), \\ f_4^{\uparrow 8} = C_{1 \times 1}(TC_{4 \times 4}(TC_{4 \times 4}(TC_{4 \times 4}(f_4)))). \end{cases} \tag{3}$$

where, $TC(\cdot)$ refers to transpose convolution, which is significant for restoring detail information, enhancing feature expression capabilities, and ensuring feature space dimensionality matching, thereby improving the accuracy and efficiency of contrastive learning. Secondly, to reduce computational costs, we use sampling operations and reshape operations to downsample the ground truth mask by a factor of four, ensuring it matches the size of $f_2^{\uparrow 2}$. The sampling operation means that if a batch contains more than 25% tampered pixels, it will be marked as tampered, and if it is less than 25%, it will be marked as real. Lastly, we utilize the semi-hard example sampling strategy Wang et al. (2021) to calculate the pixel contrastive loss. For the specific level features of each training image, we construct a memory bank by sampling 10 pixels from each class. Then, for each pixel anchor, we draw 512 closest negative samples and 512 randomly selected negative samples from the memory bank to calculate the contrastive loss. The process is as follows:

$$cl = \frac{1}{|P_z|} \sum_{z^+ \in P_z} - \log \frac{\exp(z \cdot z^+/\tau)}{\exp(z \cdot z^+/\tau) + \sum_{z^- \in N_z} \exp(z \cdot z^-/\tau)} \tag{8}$$

where, the positive sample for pixel $z$ is denoted by $P_z$, and the negative sample by $N_z$. $\tau$ represents a temperature hyper-parameter, which is set to 0.1 here. The total pixel-level contrastive loss is denoted by $L_{cl}$, $L_{cl} = cl_{f_2} + cl_{f_3} + cl_{f_4}$.

## 3.2 LOCALIZING TAMPERED REGIONS

In the Localizing Tampered Regions Stage, we designed three decoder branches: the body region decoupling branch with a Coarse Prediction Decoder (CPD), and four Body-Aware Blocks (BABs);

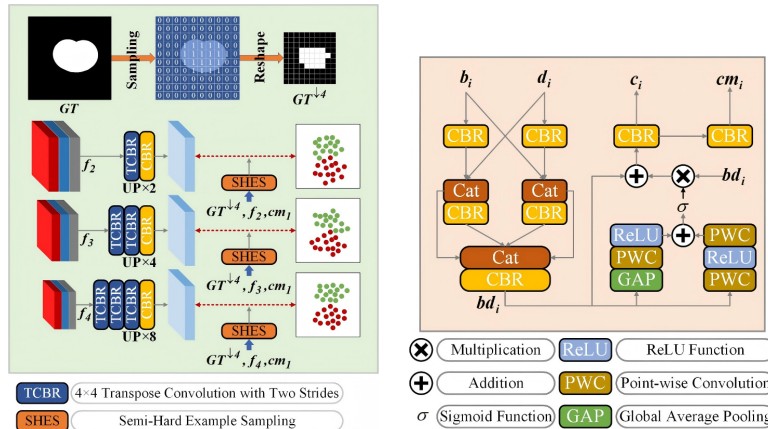

Figure 4: $(left)$ The structure of MCL. $(right)$ The structure of IFB.

the detail region decoupling branch with an Edge Prediction Module (EPM), and four Detail-Aware Blocks (DABs); and the interaction and integration branch with four Interaction Fusion Blocks (IFBs). Next, we will provide a detailed description of these key modules.

CPD is mainly used for generating coarse localization result $cp$. To ensure semantic consistency within levels and contextual connectivity between levels, we first utilize cross-connections to fully exploit the semantic context relationships across different levels, then progressively aggregate features from various levels, and finally generate the $cp$ through a $1 \times 1$ CBR.

$$\begin{cases} f_4^* = f_4 \otimes D_{\times 2}(f_3), \\ f_3^* = f_3 \otimes D_{\times 2}(f_2) \otimes U_{\times 2}(f_4) \otimes U_{\times 2}(f_4^*), \\ f_2^* = f_2 \otimes U_{\times 2}(f_3) \otimes U_{\times 2}(f_3^*). \end{cases} \tag{9}$$

$$cp = C_{1 \times 1}(C_{3 \times 3}(Cat(f_2^*, U_{\times 2}(C_{3 \times 3}(Cat(f_3^*, U_{\times 2}(f_4^*))))))). \tag{10}$$

EPM is primarily used for generating the initial predicted edge map $em$. First, EPM takes low-level feature $f_1$ and high-level feature $f_4$ as inputs, with the low-level feature containing rich edge details and the high-level feature possessing abundant semantic information. Secondly, to enhance patterns related to edges, we introduce the Sobel layer. The Sobel layer differentiates edge-related pixels from other pixels in the given feature map using edge-related weights[7]. Finally, we obtain the edge map $em$ through a series of convolutional operations.

$$em = \sigma(C_{1 \times 1}(C_{3 \times 3}(Cat(f_1 \oplus S(f_1), U_{\times 8}(f_4 \oplus S(f_4)))))). \tag{11}$$

where, $\oplus$ refers to element addition operation, $\sigma$ refers to Sigmoid function, $S(\cdot)$ refers to the Sobel operator.

DAB is primarily used to analyze the detail regions and generate detail maps. To obtain the most comprehensive information, DAB first integrates the global context feature $g^d$, the interactive fusion feature $c_{i+1}$ output from the previous level's IFB (when the current feature is not $f_4^d$), and the features $f_i^d$.

$$\begin{cases} a_4^d = C_{1 \times 1}(Cat(f_4^d, g^d)), \\ a_i^d = C_{1 \times 1}(Cat(f_i^d, g^d, c_{i+1})), \quad i = 1, 2, 3 \end{cases} \tag{14}$$

Secondly, when faced with tampering regions that perfectly blend into the background, by capturing multi-scale features, the model can better understand the shape and contour of the target. We input $a_i^d$ into a $3 \times 3$ CBR and three parallel dilated convolutions with dilation rates of 2, 3, and 4, respectively, to meticulously capture multi-scale features within different receptive fields. Then, we calculate the feature differences between adjacent scales to identify details that were not captured under different receptive fields.

$$\begin{cases} DB_n = C_{3 \times 3}(a_i^d), \quad n = 1 \\ DB_n = DC_{\text{Dilation Rate}=n}(a_i^d), \quad n = 2, 3, 4 \end{cases} \tag{15}$$

$$res_m = DB_m - DB_{m+1}, \quad m = 1, 2, 3 \tag{16}$$

where, $DC(\cdot)$ refers to dilated convolution. Finally, we fuse $a_i^d$ with all feature residuals to obtain the output features $d_i$ and the detail map $dm_i$.

$$d = C_{3\times3}(a^d \oplus res_1 \oplus res_2 \oplus res_3). \tag{17}$$

$$dm = C_{1\times1}(d). \tag{18}$$

BAB is designed to analyze the body of tampering regions, with a structure identical to DAB. Its outputs include the body map $bm_i$ and the body decoupling features $b_i$.

The main purpose of IFB is to fuse the output features of the two decoupling branches at a specific level and generate complete localization results[28]. Its structure is shown on the right of Figure 4. Specifically, feature interaction and feature fusion are first carried out through mutual connection. This process can be described as:

$$d_i^* = Cat(C_{3\times3}(d_i), b_i). \tag{19}$$

$$b_i^* = Cat(C_{3\times3}(b_i), d_i). \tag{20}$$

$$bd_i = C_{3\times3}(Cat(d_i^*, C_{3\times3}(d_i^*), b_i^*, C_{3\times3}(b_i^*))). \tag{21}$$

Next, the Multi-Scale Channel Attention (MSCA) Dai et al. (2021) is introduced to further optimize information fusion by enhancing feature selectivity. MSCA consists of two branches: one obtains global context information through global average pooling, and the other obtains local context information. This process can be described as:

$$c_i = C_{3\times3}(bd_i \otimes MSCA(bd_i) \oplus bd_i). \tag{22}$$

where, $MSCA(\cdot)$ refers to MSCA. $c_i$ refers to the interactive fusion features, which serve two purposes: firstly, it is used to predict the complete localization result $cm_i$, and secondly, it is used for the decoupling tasks in the next hierarchical level.

### 3.3 LOSS FUNCTION

We utilize a combination of WBCE and WIoU loss functions Wei et al. (2020) for precise and coarse localization maps, respectively. For body and detail maps, BCE loss is applied, while Dice loss Xie et al. (2020) is used for edge supervision.

$$L_{cm} = \sum_{i=1}^{4} L_{WBCE+WIOU}(cm_i, GT). \tag{23}$$

$$L_{cp} = L_{WBCE+WIOU}(cp, GT). \tag{24}$$

$$L_{bd} = \sum_{i=1}^{4} L_{BCE}(dm_i, GT_{dm}) + \sum_{i=1}^{4} L_{BCE}(bm_i, GT_{bm}). \tag{25}$$

$$L_{em} = L_{dice}(em, GT_{em}) \tag{26}$$

where, $GT_{dm}$, $GT_{bm}$, and $GT_{em}$ represent the detail region labels, body region labels, and edge labels, respectively. Then the total loss function is as follows:

$$L_{total} = \chi L_{cm} + \delta L_{cp} + \varepsilon L_{bd} + \phi L_{em} + \varphi L_{cl}. \tag{27}$$

where, $\chi, \delta, \varepsilon, \phi, \varphi$ are hyperparameters, we set them to 1, 1, 1, 1, 0.5 respectively.

## 4 DMI DATASET

To solidify the foundation for DMIL tasks, we introduce a high-quality dataset named DMI. It covers five mainstream tampering models, each contributing 10,000 tampered images, which significantly influence image editing and tampering. These models include BrushNet (BN) Ju et al. (2024), Paint by Example (PE) Yang et al. (2023), Inpaint Anything (IA) Yu et al. (2023),PowerPaint (PP) Zhuang et al. (2024) , and Repaint (RP) Lugmayr et al. (2022) Secondly, the DMI dataset encompasses five prevalent tampering types, representing the main trends in current image tampering techniques. These types include example-guided generative forgery, text-guided forgery, context- aware forgery, object removal, and shape-guided forgery. Lastly, the DMI dataset span various scenes and subjects, including natural landscapes, urban environments, and portraits, offering rich material for the training and testing of tampering detection algorithms.

## 4.1 Production of the DMI Dataset

To construct the DMI dataset, we initially curated original images from a variety of high-quality sources, including datasets such as Wider Person Zhang et al. (2019), Total-Text Ch'ng et al. (2020), RCTW 17 Shi et al. (2017), Coco Lin et al. (2014), CelebA-HQ Liu et al. (2015), and Place2 Zhou et al. (2017), as well as real photographs from the internet that are free of copyright restrictions, ensuring both diversity and authenticity in our dataset. Subsequently, we created tampering masks through hand drawing, SAM, or the object selection tool in Photoshop, with the specific location, target, and shape of the tampering chosen freely by the creators to simulate real-world tampering scenarios while ensuring the masks have semantic meaning. Furthermore, we manually crafted or selected prompts and examples for the diffusion model to enhance the credibility of the forged images and increase the challenge of the dataset. The generated images were then filtered, retaining only those without obvious tampering traces, with semantic meaning, and of the highest visual quality. Finally, we subjected the selected tampered images to various post-processing operations, including selective blurring, JPEG compression, JPEG restoration, downscaling, and super-resolution, to further eliminate traces of tampering and increase detection difficulty. The number of images undergoing post-processing operations is detailed in Table 1, with a typical image often undergoing multiple post-processing operations.

## 4.2 Datasets Comparison

As shown in Table 2, we conducted a comparative analysis of 18 image tampering datasets. We observed that most existing datasets primarily focus on traditional tampering types, such as splicing, copy-move, and removal, which rely mainly on manual operations or non-deep learning techniques. Forensics models trained on these datasets are no longer suitable for the current reality of AIGC image tampering scenarios. Among these datasets, only DID, AutoSplice, CoCoGlide, and DMI include deep learning-based generative models. The DID dataset mainly contains inpainting methods based on CNNs and GANs, which leave noticeable tampering traces. Although AutoSplice and CoCoGlide incorporated diffusion models, they suffer from small scale and limited tampering types. In contrast, the DMI dataset includes five mainstream diffusion model-based local generative forgery methods, covering five major tampering types, and provides 50,000 high-quality tampered images, effectively addressing the shortcomings of current mainstream datasets.

Table 1: Number of post-processed images.

| Operation | Selective blur | JPEG compression | JPEG restoration | Downsizing | Super-resolution |
|---|---|---|---|---|---|
| Number of images | 21K | 12K | 12K | 5K | 5K |

Table 2: Summary of previous image tampering datasets and our DMI.

| Datasets | Number of forged images | Traditional image forgery | | | AIGC image forgery | | | | |
|---|---|---|---|---|---|---|---|---|---|
| | | Splicing | Copy-move | Removal | Example-guided | Text-guided | Context-aware | Shape-guided | Object Removal |
| Columbia Ng et al. (2009) | 180 | ✓ | ✗ | ✓ | ✗ | ✗ | ✗ | ✗ | ✗ |
| CASIA1 Dong et al. (2013) | 921 | ✓ | ✓ | ✗ | ✗ | ✗ | ✗ | ✗ | ✗ |
| CASIA2 Dong et al. (2013) | 5,123 | ✓ | ✓ | ✗ | ✗ | ✗ | ✗ | ✗ | ✗ |
| Wild Zampoglou et al. (2015) | 9,657 | ✓ | ✓ | ✓ | ✗ | ✗ | ✗ | ✗ | ✗ |
| IMD2020 Novozamsky et al. (2020) | 35,000 | ✓ | ✓ | ✓ | ✗ | ✗ | ✗ | ✗ | ✗ |
| Nist16 NIST (2016) | 564 | ✓ | ✓ | ✓ | ✗ | ✗ | ✗ | ✗ | ✗ |
| HTSI12K Hao et al. (2024a) | 12,000 | ✓ | ✗ | ✗ | ✗ | ✗ | ✗ | ✗ | ✗ |
| IPM15K Ren et al. (2023) | 15,000 | ✓ | ✓ | ✓ | ✗ | ✗ | ✗ | ✗ | ✗ |
| TMI12K Ren et al. (2024) | 12,000 | ✓ | ✓ | ✓ | ✗ | ✗ | ✗ | ✗ | ✗ |
| MICCF-2000 Amerini et al. (2011) | 700 | ✓ | ✗ | ✓ | ✗ | ✗ | ✗ | ✗ | ✗ |
| VIPP Synth Amerini et al. (2011) | 4,800 | ✓ | ✗ | ✓ | ✗ | ✗ | ✗ | ✗ | ✗ |
| MFC2018 Guan et al. (2019) | 3,265 | ✓ | ✓ | ✓ | ✗ | ✗ | ✗ | ✗ | ✗ |
| MFC2019 Guan et al. (2019) | 5,750 | ✓ | ✓ | ✓ | ✗ | ✗ | ✗ | ✗ | ✗ |
| RLS26K Hao et al. (2024b) | 26,000 | ✓ | ✓ | ✓ | ✗ | ✗ | ✗ | ✗ | ✗ |
| DID Wu & Zhou (2021) | 10,000 | ✗ | ✗ | ✓ | ✗ | ✗ | ✓ | ✗ | ✓ |
| AutoSplice Jia et al. (2023) | 2,273 | ✗ | ✗ | ✗ | ✗ | ✓ | ✗ | ✗ | ✗ |
| CoCoGlide Guillaro et al. (2023) | 512 | ✗ | ✗ | ✗ | ✗ | ✓ | ✗ | ✗ | ✓ |
| DMI (Ours) | 50,000 | ✗ | ✗ | ✗ | ✓ | ✓ | ✓ | ✓ | ✓ |

## 5 EXPERIMENTS

### 5.1 IMPLEMENTATION DETAILS

In this paper, we implemented DMIL-Net using the PyTorch framework, and the model training is facilitated with the Adam optimizer. The learning rate is configured at $1 \times 10^{-4}$, and the batch size is set to 10. The model undergoes a total of 50 training epochs. All experiments were carried out on a single NVIDIA GeForce RTX 4090 GPU device. We resize all the input images to and augment them by randomly horizontal flipping. We choose the F1-Score (F1) to evaluate the performance of DMIL-Net. The experiments utilized a total of six datasets, including DMI, DID Wu & Zhou (2021), IMD Novozamsky et al. (2020), Nist16 NIST (2016), DEFACTO (Splicing) Mahfoudi et al. (2019), and AutoSplice (AUTO) Jia et al. (2023). Among them, DID was used solely as a test set, while DMI, IMD, Nist16, DEFACTO, and AUTO were divided into train and test sets in an 8-to-2 ratio.

### 5.2 ABLATION STUDY

We conducted a detailed ablation study on DMIL-Net. We devised six schemes, with the specific configuration of each scheme detailed in Table 3. The experiments were conducted on the DMI dataset, which includes five sub-test sets in its test set: BN, PE, IA, PP, and RP, with each sub-test set containing 2,000 tampered images. The results, as shown in Table 4, lead to the following conclusions: Firstly, the incorporation of noise views and high-frequency information significantly improved model performance, with Scheme 3 showing a 0.047 increase in F1 over Scheme 1 when dealing with the PP subset. Then, the introduction of a multi-level contrastive learning strategy enhanced multi-view feature fusion, leading to better detection of tampering regions, and Scheme 4 outperformed Scheme 3 across all subsets. Notably, there was a 0.16 increase in F1 for the IA subset and a 0.12 increase for the PP subset. Lastly, the decoupling and integration strategy preserved local details while ensuring the completeness of localization results. Scheme 6 demonstrated a clear performance improvement over Scheme 4 in all subsets.

### 5.3 COMPARISON WITH STATE-OF-THE-ART METHODS

We compared DMIL-Net with five mainstream methods: MVSS-Net Dong et al. (2022), MFI-Net Ren et al. (2023), TA-Net Shi et al. (2023), CFL-Net Niloy et al. (2023), and EMF-Net Ren et al. (2024). We selected DMI as the training and testing dataset. The results are shown in Table 5. The experimental results indicate that DMIL-Net outperforms all other methods across all subsets. EMF-Net has the second-best performance, but DMIL-Net significantly surpasses EMF-Net. The localization results of the six methods are depicted in Figure 5. DMIL-Net produces the most complete localization results with the richest edge details. This demonstrates that DMIL-Net has excellent performance in dealing with diffusion-based generative forgery operations.

### 5.4 GENERALIZATION EVALUATION

The DID dataset includes ten representative inpainting methods, six of which are deep learning-based, including GC Yu et al. (2019), CA Yu et al. (2018), SH Yan et al. (2018), EC Nazeri et al. (2019), LB Wu et al. (2021), and RN Yu et al. (2020), and four traditional methods, including NS Bertalmio et al. (2001), LR Telea (2004), PM Herling & Broll (2014), and SG Huang et al. (2014). Each of these ten methods contributed 1000 fake images. In this experiment, we trained DMIL-Net on the DMI train set and tested it against five other methods on the DID dataset. As shown in Table 7, cross-dataset testing is challenging, with few models achieving satisfactory results. DMIL-Net slightly underperformed EMF-Net in the GC subset. However, it outperformed other methods in the remaining nine subsets, demonstrating its robust generalization ability.

### 5.5 EXTENSIBILITY EVALUATION

IMD is a real-life manipulated dataset that focuses more on simulating complex and challenging real-life situations. Nist16 includes three types of tampering: splicing, copy-move, and removal. DEFACTO contains the splicing tampering type. AUTO is an AIGC dataset constructed using the

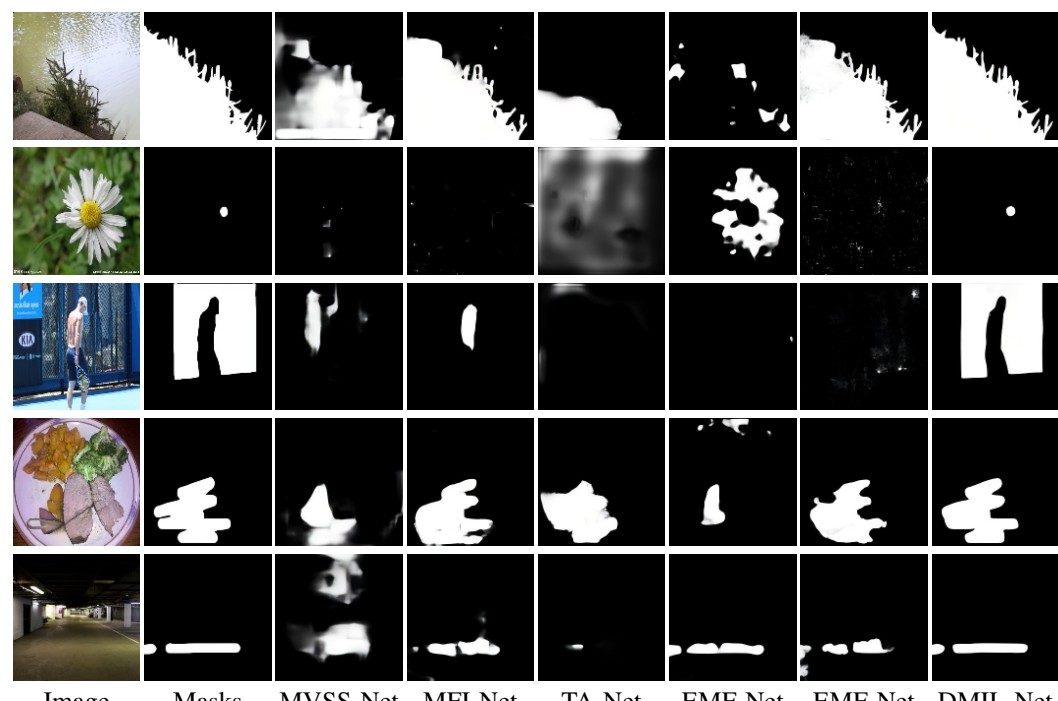

Image    Masks    MVSS-Net    MFI-Net    TA-Net    EMF-Net    EMF-Net    DMIL-Net

Figure 5: The localization results of six SOTA methods on DMI.

Table 3: Configurations of six schemes.

| Methods | Configurations |
|---|---|
| Scheme 1 | Base + None |
| Scheme 2 | Scheme 1 + HFE |
| Scheme 3 | Scheme 2 + NRE |
| Scheme 4 | Scheme 3 + MCL |
| Scheme 5 | Scheme 4 + BAB+ DAB +IFB |
| Scheme 6 | Scheme 5 + CPD+ EPM + REB + EEB |

Table 4: Results of the ablation study.

| Methods | BN | PE | IA | PP | RP |
|---|---|---|---|---|---|
| Scheme 1 | 0.885 | 0.865 | 0.848 | 0.852 | 0.586 |
| Scheme 2 | 0.887 | 0.875 | 0.849 | 0.867 | 0.593 |
| Scheme 3 | 0.889 | 0.885 | 0.857 | 0.899 | 0.604 |
| Scheme 4 | 0.891 | 0.892 | 0.873 | 0.911 | 0.609 |
| Scheme 5 | 0.906 | 0.897 | 0.891 | 0.920 | 0.618 |
| Scheme 6 | **0.918** | **0.928** | **0.893** | **0.936** | **0.627** |

DALL-E2 model for automatic image editing. In this experiment, we evaluated the extensibility of DMIL-Net on the four aforementioned datasets. As shown in Table 6, the results indicate that DMIL-Net has a significant advantage in handling tampering types other than local generative forgery, achieving the best performance across all four datasets and demonstrating its excellent extensibility.

## 5.6 ROBUSTNESS EVALUATION

In this experiment, we individually applied Gaussian noise, Gaussian filtering, gamma correction, and scaling attacks to the BN subset of the DMI test set to assess the stability of DMIL-Net. As illustrated in Figure 6, all of these attacks increased the challenge of localization, resulting in a noticeable performance decline in all models as the severity of the attacks escalated. Despite this, DMIL-Net consistently outperformed other models, indicating its high resistance to the individual effects of Gaussian noise, Gaussian filtering, gamma correction, and scaling attacks. These findings highlight DMIL-Net's robustness against a range of image perturbations, even when subjected to varying levels of each specific attack.

Table 5: Results of six SOTA methods on DMI dataset.

| Methods | BN | PE | IA | PP | RP |
|---|---|---|---|---|---|
| MVSS-Net | 0.774 | 0.689 | 0.606 | 0.717 | 0.511 |
| MFI-Net | 0.864 | 0.856 | 0.722 | 0.863 | 0.564 |
| TA-Net | 0.581 | 0.483 | 0.266 | 0.587 | 0.288 |
| CEL-Net | 0.809 | 0.719 | 0.656 | 0.791 | 0.486 |
| EMF-Net | 0.899 | 0.881 | 0.855 | 0.875 | 0.589 |
| DMIL-Net | **0.918** | **0.928** | **0.893** | **0.936** | **0.627** |

Table 6: Results of six methods on the IMD, Nist16, DEFACIO, and AUTO datasets.

| Methods | IMD | Nist16 | DEFACIO | AUTO |
|---|---|---|---|---|
| MVSS-Net | 0.363 | 0.879 | 0.715 | 0.913 |
| MFI-Net | 0.457 | 0.866 | 0.844 | 0.922 |
| TA-Net | 0.433 | 0.883 | 0.831 | 0.934 |
| CEL-Net | 0.433 | 0.884 | 0.907 | 0.917 |
| EMF-Net | 0.453 | 0.900 | 0.842 | 0.929 |
| DMIL-Net | **0.593** | **0.923** | **0.924** | **0.958** |

Table 7: The cross-dataset test results of six methods, with the train set being DMI.

| Methods | GC | CA | SH | EC | LB | RN | NS | LR | PM | SG |
|---|---|---|---|---|---|---|---|---|---|---|
| MVSS-Net | 0.261 | 0.319 | 0.539 | 0.239 | 0.288 | 0.445 | 0.380 | 0.301 | 0.224 | 0.533 |
| MFI-Net | 0.249 | 0.515 | 0.794 | 0.606 | 0.709 | 0.643 | 0.359 | 0.426 | 0.363 | 0.659 |
| TA-Net | 0.127 | 0.106 | 0.313 | 0.351 | 0.189 | 0.237 | 0.054 | 0.115 | 0.027 | 0.091 |
| CFL-Net | 0.210 | 0.186 | 0.406 | 0.176 | 0.140 | 0.347 | 0.227 | 0.138 | 0.116 | 0.293 |
| EMF-Net | **0.306** | 0.228 | 0.488 | 0.167 | 0.377 | 0.390 | 0.172 | 0.079 | 0.049 | 0.140 |
| DMIL-Net | 0.286 | **0.848** | **0.916** | **0.859** | **0.943** | **0.861** | **0.653** | **0.743** | **0.797** | **0.897** |

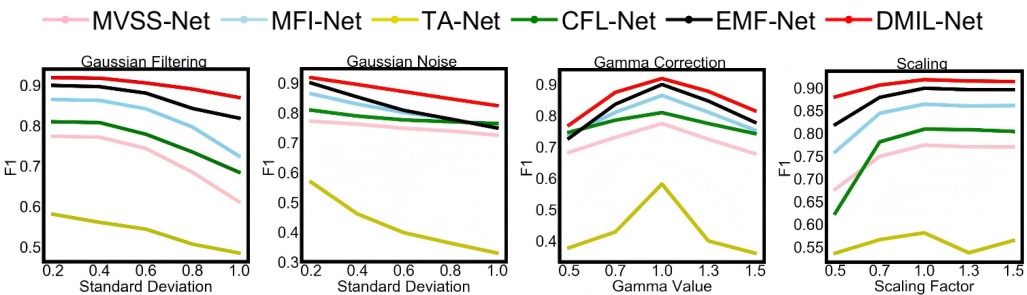

Figure 6: Results of robustness evaluation.

## 6 CONCLUSION

In this paper, we propose a novel network, DMIL-Net, for the DMIL task and construct the DMI dataset based on five different diffusion models. DMIL-Net incorporates the multi-view feature learning strategy, and the tampering region decoupling and integration strategy. Extensive experiments have proven the effectiveness of the proposed strategies and demonstrated DMIL-Net's significant advantages in localization performance, generalization, extensibility, and robustness. In future work, for the DMIL task, we consider designing new forensic models based on diffusion models to further enhance the generalization capabilities against the latest generation model image tampering tasks.

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

## A  GENERALIZATION EVALUATION VISUALIZATION

In generalization evaluation experiment, the localization results of cross-dataset testing are shown in Figure 7, demonstrate that DMIL-Net can generate more precise localization results compared to other methods, confirming the superiority of DMIL-Net in localization performance.

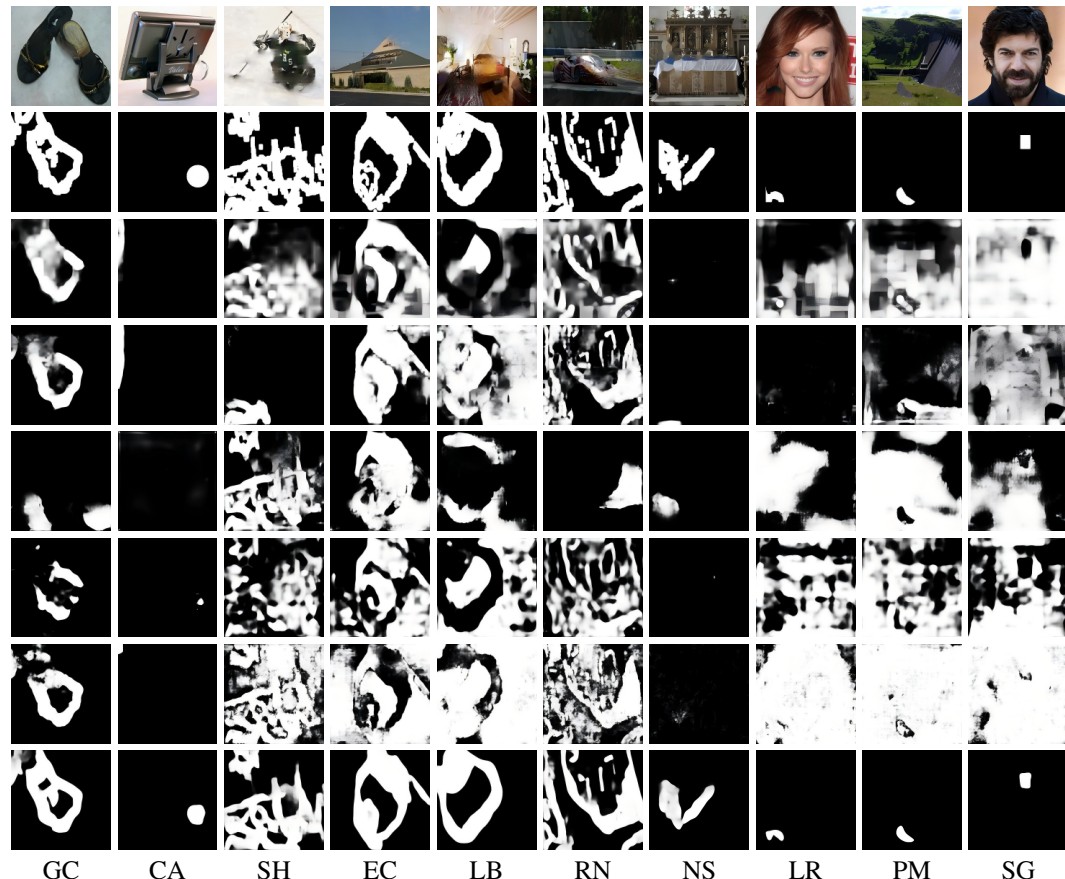

Figure 7: The localization results of cross-dataset testing. The first to eighth rows correspond to fake images, masks, MVSS-Net, MFI-Net, TA-Net, CFL-Net, EMF-Net, and DMIL-Net.

