# OpenReview forum: "DMIL-Net: A Multi-View Fusion and Region Decoupling Network For Diffusion-Based Generative Image Forgery Localization"
_ICLR.cc/2026/Conference — Submitted to ICLR 2026_

### Official Review · Reviewer_8n7g · 2025-10-20

**Soundness:** 3
**Presentation:** 1
**Contribution:** 2
**Rating:** 4
**Confidence:** 4

**Summary:**

This paper introduces DMIL-Net, a new framework for diffusion-based generative image forgery localization (DMIL). The model leverages three key strategies:

- Multi-view feature learning, integrating RGB, noise, and high-frequency information to capture tampering cues.

- Multi-level contrastive learning, enhancing long-range dependencies across modalities and improving feature fusion.

- Region decoupling and integration, which separately models body and detail regions for more accurate localization.

In addition, the authors introduce a new dataset, DMI, comprising 50,000 forged images generated by five diffusion-based inpainting methods with various post-processing operations. Experimental results show that DMIL-Net surpasses existing SOTA methods in localization accuracy, generalization, extensibility, and robustness.

**Strengths:**

- Comprehensive architecture: The proposed combination of multi-view learning and multi-level contrastive fusion is well-motivated and carefully designed.

- Thorough experiments: Extensive evaluations are provided, including ablation, cross-dataset generalization, extensibility to classical datasets, and robustness tests under noise and compression attacks.

**Weaknesses:**

- The Method section is somewhat complex, with numerous formulas and detailed information, making it difficult to read. The lack of detailed descriptions of each module's functionality (including the problems it addresses and how they are solved) makes it difficult to understand the innovations of DMIL-Net.

- The extraction and learning of multi-view and multi-level features have been covered in other papers (e.g., MVSS-Net[1], MMFusion[2]). However, it seems that the authors do not show how to better utilize these features to localize tampering in diffusion-based generation images, which also weakens the innovation of the article.

- The DMI Dataset, one of the article's innovations, doesn't quite fit the description in the Experiment section. Secondly, DMI simply aggregates existing image editing models to construct it, making it seem like an engineering exercise rather than an innovation.

- In ablation experiments, the improvement in localization performance across various modules is unclear. For example, the effect of NRE can only be seen by comparing Scheme 2 and Scheme 3, but it's unclear whether this improvement is related to the presence of HFE. It would be better to add NRE directly to Scheme 1 and conduct a comparison. Similar experimental verification may also be required for other modules.

Overall: The main problem with the article is that the Method section is overly complex, lacking the necessary motivations and key design details for each module. The excessive detail obscures the core innovations. Furthermore, the overall structure of the article is somewhat unbalanced. For example, the DMI Dataset, one of the innovations, is only briefly described in the Experiment section.

Minor Issues:

- Line 174, $r_i,n_i,n_i$ should be corrected to $r_i,h_i,n_i$.

[1] MVSS-Net: Multi-View Multi-Scale Supervised Networks for Image Manipulation Detection

[2] Exploring Multi-Modal Fusion for Image Manipulation Detection and Localization

**Questions:**

Please refer to the Weaknesses.

---

### Official Review · Reviewer_qXSa · 2025-10-28

**Soundness:** 2
**Presentation:** 2
**Contribution:** 2
**Rating:** 4
**Confidence:** 5

**Summary:**

The paper addresses the challenge of detecting and localizing tampered regions in images generated by diffusion-based models, which are increasingly difficult to distinguish due to their high fidelity. The authors propose DMIL-Net, a novel framework that integrates multi-view feature learning (combining RGB, noise, and high-frequency domains), multi-level contrastive learning for cross-modal dependency modeling, and a region decoupling strategy that separates tampered areas into body and edge components for iterative refinement. A key contribution is the construction of the DMI dataset, containing 50,000 synthetic forgery images generated via five prevalent diffusion methods, annotated with diverse tampering patterns including object removal, text-guided editing, and shape-aware manipulation. Extensive experiments demonstrate superior performance over existing methods in localization accuracy, robustness to distribution shifts, and generalization across different tampering scenarios, with ablation studies validating the effectiveness of each architectural component.

**Strengths:**

Originality
The paper demonstrates strong originality through both methodological innovation and dataset contribution. The proposed DMIL-Net framework creatively combines multi-view feature learning (RGB, noise, and high-frequency domains) with multi-level contrastive learning to model cross-modal dependencies, which is a novel approach for diffusion-based forgery detection. The region decoupling strategy—iteratively refining tampered areas by separating body and edge components—is a new perspective that addresses the inherent challenges of diffusion-generated forgeries. Additionally, the construction of the DMI dataset (50,000 synthetic images generated via five diffusion methods, annotated with diverse tampering patterns) is an original contribution, as existing datasets focus on traditional forgeries (e.g., copy-paste, splicing) but lack coverage of diffusion-specific manipulations.

Quality:
The experimental validation is thorough and methodologically sound. The authors compare DMIL-Net against state-of-the-art methods across multiple metrics (e.g., localization accuracy, robustness to distribution shifts) and demonstrate consistent improvements. Ablation studies meticulously disentangle the contributions of individual components (e.g., multi-view features, contrastive learning, region decoupling), providing strong empirical support for their design choices. The evaluation includes diverse tampering scenarios (object removal, text-guided editing, shape-aware manipulation) and tests generalization across unseen diffusion models, which strengthens the credibility of the claims. The technical depth and reproducibility are further enhanced by detailed implementation details and public dataset availability.

Clarity:
The problem formulation is clearly articulated in the introduction, with a concise overview of limitations in prior work. The methodology section logically unfolds the design of DMIL-Net, using intuitive diagrams and pseudocode to explain complex components like the multi-level contrastive module and region decoupling. The experiments are presented in a structured manner, with comparisons to baselines and qualitative results that effectively highlight the method’s strengths. Technical terms are well-defined, and the language is precise without being overly verbose, making the paper accessible to both domain experts and broader machine learning audiences.

Significance:
The work addresses a highly relevant and timely problem: detecting forgeries generated by diffusion models, which are rapidly becoming the dominant paradigm in image synthesis. As diffusion-based manipulations grow in sophistication and prevalence, the ability to detect and localize tampered regions is critical for applications in security, forensics, and media integrity. The proposed framework not only advances the technical capabilities of forgery detection but also provides the community with a benchmark dataset that will catalyze future research.

**Weaknesses:**

1. Limited Evaluation on Real-World Data and Generalization to Unseen Manipulation Types
While the DMI dataset is comprehensive for diffusion-based forgeries, it is entirely synthetic. The experiments focus exclusively on this synthetic data, leaving open critical questions about the method’s performance on real-world tampered images (e.g., from social media or forensic casework). Additionally, the synthetic forgeries in DMI are generated via only five diffusion methods and three tampering patterns (object removal, text-guided editing, shape-aware manipulation). Real-world scenarios may involve hybrid manipulations (e.g., combining object removal with style transfer) or novel diffusion variants not included in the training set. The paper does not address how DMIL-Net might generalize to such cases, nor does it provide results on existing real-world forensic datasets (e.g., Columbia or MICC datasets adapted for diffusion forgeries).

2. Over-Reliance on Multi-View Features Without Ablation on Computational Cost
The multi-view feature learning (RGB, noise, high-frequency domains) is a key strength, but the paper does not analyze the computational trade-off between accuracy and inference speed. For instance, the high-frequency domain extraction (e.g., Laplacian filtering) adds preprocessing steps that may be computationally expensive in deployment. Furthermore, the ablation study focuses only on feature contribution, not on whether the added complexity is justified for real-time applications.

3. Lack of Theoretical Justification for Region Decoupling Strategy
The region decoupling strategy (separating tampered areas into body and edge components for iterative refinement) is empirically effective but lacks a theoretical foundation. The paper does not explain why separating these components improves performance—does the decoupling reduce intra-class variability in tampered regions? Does it mitigate boundary ambiguity in diffusion-generated forgeries? Without a principled explanation, the approach risks being seen as an ad hoc heuristic rather than a generalizable framework.

4. Incomplete Comparison to Non-Diffusion-Based Forgery Detection Methods
While the paper compares DMIL-Net to state-of-the-art diffusion-specific methods, it does not benchmark against traditional forgery detection approaches (e.g., image splicing detection or GAN-based forgery localization). This omission is significant because some traditional methods (e.g., detecting noise inconsistencies or illumination artifacts) might still perform reasonably well on diffusion forgeries, especially for simpler tampering types like object removal.

5. Ambiguity in "Tampered Edge" Definition and Annotation Consistency
The DMI dataset annotations define "tampered edges" as boundaries between forged and authentic regions, but the paper does not specify how these edges were annotated (e.g., manually by experts, automated post-processing of diffusion outputs). If annotations are noisy or inconsistent (e.g., blurry edges or misaligned masks), this could bias the evaluation of the region decoupling strategy.

6. Limited Discussion on Adversarial Robustness
The paper emphasizes robustness to distribution shifts but does not test whether DMIL-Net is vulnerable to adversarial attacks (e.g., small perturbations designed to fool forgery detectors). Given the adversarial nature of diffusion-based forgeries (e.g., attackers may optimize for stealthiness), this is a critical gap. Improvement: Evaluate the model under adversarial conditions (e.g., FGSM or PGD attacks) and propose mitigation strategies (e.g., adversarial training or input purification).

**Questions:**

1. Real-World Generalization and Unseen Tampering Types
The experiments focus entirely on synthetic data from five diffusion methods and three tampering patterns. How does DMIL-Net perform on real-world tampered images (e.g., social media or forensic datasets) or hybrid/unknown manipulation types (e.g., diffusion-based forgeries combined with traditional splicing)?

2. Computational Cost of Multi-View Features
The multi-view feature learning (RGB, noise, high-frequency domains) improves accuracy but adds preprocessing steps (e.g., Laplacian filtering). What is the trade-off between accuracy gains and inference speed/memory usage compared to simpler baselines (e.g., using only RGB features)?

3. Theoretical Justification for Region Decoupling
The region decoupling strategy (body-edge separation) is empirically effective, but the paper lacks a theoretical explanation for why this approach improves localization. Is there a principled connection between diffusion model properties and the decoupling mechanism?

4. Comparison to Traditional Forgery Detection Methods
The paper compares DMIL-Net to diffusion-specific methods but omits traditional forgery detection techniques (e.g., noise inconsistency or illumination-based methods). Are these methods competitive on diffusion forgeries, and if not, why?

5. Annotation Consistency in the DMI Dataset
The DMI dataset defines "tampered edges" but does not clarify how annotations were created (e.g., manual vs. automated). How consistent are the edge annotations across annotators or diffusion methods?

6. Adversarial Robustness of DMIL-Net
The paper claims robustness to distribution shifts but does not test adversarial attacks (e.g., FGSM or PGD). Could diffusion-based forgeries be optimized to evade DMIL-Net?

7. Interpretability of Multi-Level Contrastive Learning
The multi-level contrastive learning module models cross-modal dependencies but does not explain how these dependencies correlate with tampering artifacts. Can the authors provide visualizations or feature maps to interpret what the module learns?

---

### Official Review · Reviewer_KvHj · 2025-10-30

**Soundness:** 2
**Presentation:** 2
**Contribution:** 2
**Rating:** 2
**Confidence:** 4

**Summary:**

This manuscript focuses on diffusion-based generative image forgery localization and introduces a framework called DMIL-Net. The proposed model integrates multi-view feature learning, multi-level contrastive learning, and a region decoupling–integration strategy to achieve accurate and detailed forgery localization. In addition, the authors construct a DMI dataset to support model training and evaluation.

**Strengths:**

The task of diffusion-based generative image forgery localization is novel and meaningful, and the constructed DMI dataset provides support and benchmarking resources for future research.

**Weaknesses:**

**Limited methodological innovation.** The core designs of the DMIL-Net framework, i.e., multi-view feature learning, multi-level contrastive learning, and region decoupling and integration, are relatively common design choices in existing image forgery detection and localization (IFDL) research. While the combination is reasonable in diffusion-based generative image forgery localization, the methodological novelty is moderate.

**The description of DMI dataset is insufficient.** Although the DMI dataset is presented as an important contribution, this manuscript provides limited detail and lacks qualitative examples or analyses, which makes it difficult to fully understand its characteristics and significance.

**The experimental evaluation needs to be more comprehensive.** Only the F1 score is reported, while key metrics such as AUC and IoU, commonly used in IFDL tasks, are missing. This makes the evaluation less comprehensive.

**Questions:**

How does DMIL-Net perform at the image level? Does it tend to produce a high false positive rate?

---

### Official Review · Reviewer_7GmT · 2025-10-31

**Soundness:** 2
**Presentation:** 1
**Contribution:** 1
**Rating:** 2
**Confidence:** 5

**Summary:**

The paper introduces DMIL-Net, a multi-view fusion and region decoupling network designed for diffusion-based generative image forgery localization (DMIL).
The approach first captures tampering traces through three complementary views: RGB, noise residual, and high-frequency streams. Then, a multi-level contrastive learning (MCL) strategy is employed to align these feature spaces. Finally, a region decoupling and integration framework separates tampered regions into “body” and “detail” components via multiple decoder branches.
To support this task, the authors also propose a new DMI dataset with 50,000 forged images generated by five diffusion-based inpainting and editing models (BrushNet, Paint-by-Example, Inpaint Anything, PowerPaint, and Repaint).
Extensive experiments on DMI and other benchmark datasets are presented, showing improvements over several existing methods.

**Strengths:**

1. The work focuses on the detection of diffusion-based forgeries, an emerging challenge in multimedia forensics.
2. The proposed DMI dataset is large-scale and diverse, covering multiple diffusion models and manipulation types. If publicly released, it could serve as a useful benchmark for the community.
3. The paper incorporates RGB, noise, and frequency information, which is intuitively reasonable and might improve the model’s ability to detect subtle generative inconsistencies.
4. The paper presents ablation studies, cross-dataset testing, and robustness experiments, suggesting careful empirical evaluation.

**Weaknesses:**

1. The proposed framework has limited novelty and conceptual contribution, which mainly integrates existing components, such as multi-view fusion, contrastive learning, and multi-branch decoders, that are well-known in image forensics, without introducing a brand new insight or distinctive mechanism.
2. In Section 3, the Capturing Tamper Traces module follows standard noise/frequency extraction strategies widely used in prior work (e.g., MVSS-Net, EMF-Net, Mesorch, IML-ViT).
3. The Multi-Level Contrastive Learning resembles existing pixel-level contrastive schemes from CFL-Net (WACV 2023) and MPC[1], which makes me believe that there's limited contribution, from my point of view.
4. The Region Decoupling and Integration mechanism is highly similar to hierarchical decoders in MVSS-Net or coarse-to-fine refinement modules in TruFor.
5. It's a heavy problem that the framework is an overly complicated and poorly motivated architecture. For example, the paper includes a large number of modules (NRE, HFE, BAB, DAB, IFB, CPD, EPM, REB, EEB), resulting in a very heavy and difficult-to-understand design. More importantly, the motivation for each level of modular granularity is unclear, and the paper lacks discussion on why each component is necessary or how they interact synergistically.
6. Figure 2 references modules such as the Region Enhancement Block (REB) that are never described in the main text. Moreover, the paper states that BAB is structurally identical to DAB, but then uses different names without explaining their distinction or specific functionality.
7. At line 199, the paper mentions a semi-hard example sampling strategy (SHES) and even shows it in Figure 4, but no textual explanation or citation is provided.
8. At line 202, a fixed configuration of 512 negative samples + 10 positive samples per class is adopted for constructing the memory bank. However, no motivation or sensitivity analysis is offered. Considering that forgery regions vary greatly in size, a fixed sampling ratio may bias the representation. Without justification, these design choices appear arbitrary and may limit reproducibility.
9. At line 289, there is an invalid or unresolved reference [28], which should be corrected. Besides, several formulae are overloaded and lack intermediate explanations, making it difficult to follow the data flow.
10. There is no analysis of computational efficiency, parameter count, or inference time, making the method difficult to reproduce or compare fairly.
11. The paper presents an incomplete comparison with recent state-of-the-art methods. The comparison excludes several relevant and competitive models that directly address AI-based or diffusion forgeries, such as CAT-Net, TruFor (CVPR 2023), IML-ViT (AAAI 2024), Mesorch [2] (AAAI 2025), and SparseViT[3](AAAI 2025). Without these, it isn't easy to assess the real advancement of DMIL-Net relative to the current state of the art. The claim of superiority is therefore not fully convincing.
12. In experiments, Table 1 and Table 2 show extremely small gains between configurations. Such marginal improvements do not convincingly demonstrate the necessity or effectiveness of each proposed block.
13. The DMI dataset creation process is not well-documented. The specific prompts, diffusion model versions (e.g., Stable Diffusion 1.5 vs SDXL), and hyperparameters are not stated and not included in the supplementary, thus I believe the dataset description lacks transparency and reproducibility.

Overall, I believe its technical novelty is limited, and the architecture is over-complicated and poorly justified. Several submodules (e.g., REB, SHES) are mentioned but never explained, and the ablation improvements are too minor to substantiate the effectiveness of the design. The experimental comparison lacks major recent related baselines, and the dataset description is insufficient for reproducibility.
Overall, the work represents an incremental combination of existing ideas rather than a new advance, falling short of the standards required for acceptance at a top-tier venue.

[1] Lou, Zijie, et al. "Exploring multi-view pixel contrast for general and robust image forgery localization." IEEE Transactions on Information Forensics and Security (2025).
[2] Zhu, Xuekang, et al. "Mesoscopic insights: orchestrating multi-scale & hybrid architecture for image manipulation localization." Proceedings of the AAAI Conference on Artificial Intelligence. Vol. 39. No. 10. 2025.
[3] Su, Lei, et al. "Can we get rid of handcrafted feature extractors? sparsevit: Nonsemantics-centered, parameter-efficient image manipulation localization through spare-coding transformer." Proceedings of the AAAI Conference on Artificial Intelligence. Vol. 39. No. 7. 2025.

**Questions:**

1. Why are some submodules like REB not described, while BAB and DAB are said to be identical? What is their functional distinction?
2. Could you explain the rationale for using a fixed 512/10 sampling in the memory bank? Did you test variable ratios depending on the forgery size?
3. What exactly is the “semi-hard example sampling strategy” (SHES) in Figure 4? Please describe it explicitly.
4. Please correct the missing citation [28] at line 289 and verify all other references.
5. How do the small performance increments in Tables 1 and 2 support the claimed effectiveness of each module? I'm not sure such a heavy design is helpful. Can you report confidence intervals or variance?
6. Why are key SOTA methods (CAT-Net, TruFor, IML-ViT, SparseViT) omitted from comparison?
7. Will the DMI dataset, including prompt texts and model versions, be released for public use?

---

### Author Response · Authors · 2025-12-04
**Summary of Questions Answered to AC**

### A. Summary of Weaknesses

1. Issues Related to the Novelty of Paper Architecture, Module Motivation, and Complexity of Model Design

R1W1 The proposed framework has limited novelty and conceptual contribution, which mainly integrates existing components, such as multi-view fusion, contrastive learning, and multi-branch decoders, that are well-known in image forensics, without introducing a brand new insight or distinctive mechanism.

R2W1 Limited methodological innovation. The core designs of the DMIL-Net framework, i.e., multi-view feature learning, multi-level contrastive learning, and region decoupling and integration, are relatively common design choices in existing image forgery detection and localization (IFDL) research. While the combination is reasonable in diffusion-based generative image forgery localization, the methodological novelty is moderate.
R1W5 It's a heavy problem that the framework is an overly complicated and poorly motivated architecture. For example, the paper includes a large number of modules (NRE, HFE, BAB, DAB, IFB, CPD, EPM, REB, EEB), resulting in a very heavy and difficult-to-understand design. More importantly, the motivation for each level of modular granularity is unclear, and the paper lacks discussion on why each component is necessary or how they interact synergistically.

R4W1 The Method section is somewhat complex, with numerous formulas and detailed information, making it difficult to read. The lack of detailed descriptions of each module's functionality (including the problems it addresses and how they are solved) makes it difficult to understand the innovations of DMIL-Net.

**Response**:
We appreciate the reviewers for raising these issues. We did not clearly articulate the innovations of this paper in the original manuscript. Compared with traditional tampering methods and GAN-based approaches, diffusion-model-generated forged images are much closer to natural images in terms of underlying statistical features and high-level semantic fusion, making this task more challenging. Our core innovation lies in the design of a localization model tailored to the unique artifact characteristics of diffusion models, rather than a simple stacking of components. Meanwhile, we have removed redundant modules to streamline the model architecture, clarified and strengthened the motivation of each core module, and designed a multi-view adaptive weight fusion mechanism to enhance the core modules, thereby highlighting the innovations of this paper.

(1) We acknowledge that multi-view fusion and multi-branch decoding are common components in the field of image forensics. However, the innovation of this study is not a simple assembly of existing components, but rather the construction of a dataset and the design of a dedicated forgery localization model to address three core challenges of local image forgery by diffusion models: faint traces, indistinct boundaries, and the lack of targeted existing datasets.

(2) The model includes two core innovative modules:
1) **Multi-view feature extraction**: Aiming at the different artifact features exhibited by diffusion-model-generated forged images across multiple modalities, it captures generated tampering traces from three dimensions: RGB, noise, and frequency domains. To address the abnormal fusion boundaries between generated content and real backgrounds typically caused by local generation in diffusion models, we adopt the Laplacian operator to obtain high-frequency views. In response to the denoising residual artifacts left in images by the iterative denoising mechanism of diffusion models, we use non-local means filtering to acquire noise residual views.
2) **Subject-detail decoupled localization**: Local generation by diffusion models usually involves the fusion process of generated content and real backgrounds, leading to inherent differences in artifact features between the internal area of the generated content and the edge area. Therefore, we decompose the tampering mask into two complementary parts: the body part (the interior of the generated region) and the detail part (the edge of the generated region and an area extending several pixels outward). Two decoupled branches are used to learn these two parts, and they are integrated to achieve more accurate localization.

---

### Author Response · Authors · 2025-12-04
**Summary of Questions Answered to AC**

(3) Additionally, we add a **multi-view adaptive weight fusion mechanism**: This mechanism connects the two core modules and accounts for the varying responses of different views to artifacts in the Body and Detail regions—the RGB view has a stronger response in the Body region, while the noise and frequency-domain views perform better in the Detail region. Instead of fusing the three views equally, we design an adaptive fusion mechanism that includes intra-view channel attention and inter-view weight learning to separately learn the optimal view fusion weights for the Body and Detail branches.

(4) The synergy of the three modules forms a complete end-to-end localization pipeline: multi-view feature extraction provides the network with rich multi-modal feature representations; the multi-view adaptive weight fusion mechanism adaptively selects the most relevant view combinations for the Body and Detail branches; subject-detail decoupled localization uses the fused features to process the subject and detail regions separately, and finally achieves collaborative localization through integration. This design enables the network to fully leverage the distinct features of diffusion-model-generated tampering across different views and regions, realizing end-to-end optimization from feature extraction to localization.

2. Issues Related to the Design of the Two Core Modules: Multi-view Feature Extraction and Region Decoupling and Integration

R1W2 In Section 3, the Capturing Tamper Traces module follows standard noise/frequency extraction strategies widely used in prior work (e.g., MVSS-Net, EMF-Net, Mesorch, IML-ViT).

R1W4 The Region Decoupling and Integration mechanism is highly similar to hierarchical decoders in MVSS-Net or coarse-to-fine refinement modules in TruFor.

R4Q2 The extraction and learning of multi-view and multi-level features have been covered in other papers (e.g., MVSS-Net[1], MMFusion[2]). However, it seems that the authors do not show how to better utilize these features to localize tampering in diffusion-based generation images, which also weakens the innovation of the article.

R3W3 Lack of Theoretical Justification for Region Decoupling Strategy The region decoupling strategy (separating tampered areas into body and edge components for iterative refinement) is empirically effective but lacks a theoretical foundation. The paper does not explain why separating these components improves performance—does the decoupling reduce intra-class variability in tampered regions? Does it mitigate boundary ambiguity in diffusion-generated forgeries? Without a principled explanation, the approach risks being seen as an ad hoc heuristic rather than a generalizable framework.

**Response**:
We appreciate the reviewers for pointing out these details, but it should be clarified that the strategies in the compared methods are all designed for traditional artifact features and are not applicable to the tampering artifact features of diffusion models. However, our view feature extraction and region decoupling and integration modules do not directly adopt standard strategies but are tailored to the generation characteristics of diffusion models. We elaborate on the two modules separately below. First, we clarify the module motivation and its theoretical basis. Then, we point out the significant differences and innovations from the strategies adopted in other papers to demonstrate the pertinence and rationality of our strategies for the diffusion forgery localization task.

#### 2.1 For View Feature Extraction
(1) **Motivation and Theoretical Basis**:
1) **Targeting edge fusion artifacts**: Local generation by diffusion models typically causes abnormal fusion boundaries between generated content and real backgrounds. We use the Laplacian operator to extract high-frequency features and capture edge features at different scales through multi-scale Gaussian smoothing, thus effectively responding to the edge fusion artifacts unique to diffusion models.
2) **Targeting denoising residual artifacts**: The iterative denoising mechanism of diffusion models leaves denoising residual artifacts in images. We apply non-local means filtering to the image and subtract the denoised result from the original image to obtain and enhance noise residuals. This noise view can retain the denoising artifacts from the diffusion generation process.

---

### Author Response · Authors · 2025-12-04
**Summary of Questions Answered to AC**

(2) **Differences from Compared Strategies and Reasons for Not Directly Adopting Previous Work**:
1) MMFusion and MVSS use BayarConv, which adaptively extracts high-frequency features via learnable high-pass filters. In contrast, we adopt a hybrid strategy combining handcrafted feature extraction and deep network learning: these handcrafted features have clear physical meanings, where the noise view reflects denoising residuals and the high-frequency view captures edge artifacts. Compared with MVSS's method, our approach offers better interpretability and stronger pertinence.
2) MMFusion and EMF-Net use SRM filters to extract high-frequency noise features, which are designed for traditional forgery traces and lack sufficient targeting for the specific artifacts of diffusion-model-generated images. SRM filters reveal statistical anomalies left by image processing operations by calculating residuals between pixel values and their neighboring pixels, but they cannot effectively capture the denoising residuals and edge fusion artifacts unique to diffusion models.
3) Noiseprint++ used in MMFusion and TruFor is a noise extractor obtained through self-supervised contrastive learning on large-scale real images (including 1,475 camera models and 24,000 images). Noiseprint++ requires separate training with high training costs. In contrast, our strategy has the advantage of achieving efficient extraction tailored to the characteristics of diffusion-model-forged images without additional training costs.
4) Mesorch uses DCT to extract high-frequency information in images. DCT employs block processing and mainly focuses on the overall frequency distribution, making it unsuitable for localizing edge fusion artifacts.
5) The IML-ViT model directly learns feature representations of tampering traces through the self-attention mechanism of ViT without using noise/frequency extraction strategies.

#### 2.2 For Region Decoupling and Integration
(1) **Motivation and Theoretical Basis**:
Local generation by diffusion models usually involves the fusion of generated content and real backgrounds, resulting in inherent differences in artifact features between the internal area of the generated content and the edge area. Meanwhile, for edge supervision commonly used in image tampering detection tasks, the number of positive edge pixels in the loss function is extremely small, while the number of non-edge negative pixels adjacent to them is very large. Treating each pixel on the edge and non-edge pixels near the edge equally leads to large prediction errors for pixels near the edge.

In summary, instead of treating the tampering mask as a whole, we decouple it into two functionally complementary parts: the body part (the interior of the tampered region) and the detail part (the edge and an area extending several pixels outward). These two parts are learned independently via dual-branch decoupling and integrated collaboratively to achieve more accurate localization.

(2) **Differences from Compared Strategies and Innovations**:
1) MVSS-Net and MMFusion use similar multi-scale features in their decoders. To streamline the model, we have removed multi-scale feature decoding from the region decoupling and integration mechanism.
2) The region decoupling and integration mechanism is essentially different from the edge supervision methods commonly used in traditional methods such as MVSS-Net and EMF-Net. Traditional edge supervision methods can improve detection accuracy by focusing on the boundary information of tampered regions, but they are not applicable to the artifact features of diffusion models. The iterative generation process of diffusion models makes the tampered edge regions highly fused with the background. Therefore, we decompose the tampering mask into body and detail parts, learn them separately, and then fuse the features to achieve precise localization.
3) **Accounting for varying view responses to Body and Detail regions**: The RGB view has a stronger response in the Body region, while the noise and frequency-domain views perform better in the Detail region. We add a multi-view adaptive weight fusion mechanism, which includes intra-view channel attention and inter-view weight learning to separately learn the optimal view fusion weights for the Body and Detail branches.

3. Issues Related to the Definition of "Tampered Edge"

R3W5 Ambiguity in "Tampered Edge" Definition and Annotation Consistency The DMI dataset annotations define "tampered edges" as boundaries between forged and authentic regions, but the paper does not specify how these edges were annotated (e.g., manually by experts, automated post-processing of diffusion outputs). If annotations are noisy or inconsistent (e.g., blurry edges or misaligned masks), this could bias the evaluation of the region decoupling strategy.

---

### Author Response · Authors · 2025-12-04
**Summary of Questions Answered to AC**

(2) **Differences from Compared Strategies and Reasons for Not Directly Adopting Previous Work**:
1) MMFusion and MVSS use BayarConv, which adaptively extracts high-frequency features via learnable high-pass filters. In contrast, we adopt a hybrid strategy combining handcrafted feature extraction and deep network learning: these handcrafted features have clear physical meanings, where the noise view reflects denoising residuals and the high-frequency view captures edge artifacts. Compared with MVSS's method, our approach offers better interpretability and stronger pertinence.
2) MMFusion and EMF-Net use SRM filters to extract high-frequency noise features, which are designed for traditional forgery traces and lack sufficient targeting for the specific artifacts of diffusion-model-generated images. SRM filters reveal statistical anomalies left by image processing operations by calculating residuals between pixel values and their neighboring pixels, but they cannot effectively capture the denoising residuals and edge fusion artifacts unique to diffusion models.
3) Noiseprint++ used in MMFusion and TruFor is a noise extractor obtained through self-supervised contrastive learning on large-scale real images (including 1,475 camera models and 24,000 images). Noiseprint++ requires separate training with high training costs. In contrast, our strategy has the advantage of achieving efficient extraction tailored to the characteristics of diffusion-model-forged images without additional training costs.
4) Mesorch uses DCT to extract high-frequency information in images. DCT employs block processing and mainly focuses on the overall frequency distribution, making it unsuitable for localizing edge fusion artifacts.
5) The IML-ViT model directly learns feature representations of tampering traces through the self-attention mechanism of ViT without using noise/frequency extraction strategies.

#### 2.2 For Region Decoupling and Integration
(1) **Motivation and Theoretical Basis**:
Local generation by diffusion models usually involves the fusion of generated content and real backgrounds, resulting in inherent differences in artifact features between the internal area of the generated content and the edge area. Meanwhile, for edge supervision commonly used in image tampering detection tasks, the number of positive edge pixels in the loss function is extremely small, while the number of non-edge negative pixels adjacent to them is very large. Treating each pixel on the edge and non-edge pixels near the edge equally leads to large prediction errors for pixels near the edge.

In summary, instead of treating the tampering mask as a whole, we decouple it into two functionally complementary parts: the body part (the interior of the tampered region) and the detail part (the edge and an area extending several pixels outward). These two parts are learned independently via dual-branch decoupling and integrated collaboratively to achieve more accurate localization.

(2) **Differences from Compared Strategies and Innovations**:
1) MVSS-Net and MMFusion use similar multi-scale features in their decoders. To streamline the model, we have removed multi-scale feature decoding from the region decoupling and integration mechanism.
2) The region decoupling and integration mechanism is essentially different from the edge supervision methods commonly used in traditional methods such as MVSS-Net and EMF-Net. Traditional edge supervision methods can improve detection accuracy by focusing on the boundary information of tampered regions, but they are not applicable to the artifact features of diffusion models. The iterative generation process of diffusion models makes the tampered edge regions highly fused with the background. Therefore, we decompose the tampering mask into body and detail parts, learn them separately, and then fuse the features to achieve precise localization.
3) **Accounting for varying view responses to Body and Detail regions**: The RGB view has a stronger response in the Body region, while the noise and frequency-domain views perform better in the Detail region. We add a multi-view adaptive weight fusion mechanism, which includes intra-view channel attention and inter-view weight learning to separately learn the optimal view fusion weights for the Body and Detail branches.

---

### Author Response · Authors · 2025-12-04
**Summary of Questions Answered to AC**

3. Issues Related to the Definition of "Tampered Edge"

R3W5 Ambiguity in "Tampered Edge" Definition and Annotation Consistency The DMI dataset annotations define "tampered edges" as boundaries between forged and authentic regions, but the paper does not specify how these edges were annotated (e.g., manually by experts, automated post-processing of diffusion outputs). If annotations are noisy or inconsistent (e.g., blurry edges or misaligned masks), this could bias the evaluation of the region decoupling strategy.

**Response**:
Reviewers may have some confusion about the production process of our dataset. We will elaborate on the dataset construction process and the generation method of body-detail labels to prove that the reviewers' concerns are unfounded.

(1) In the dataset production process, we first create masks of tampered regions using hand-drawing, SAM, or Photoshop tools. Only after obtaining clearly defined masks do we use diffusion models to generate forged content within these mask regions based on prompts. For each generated tampered image, regardless of the diffusion model method used for forgery, the same ground-truth mask is used for evaluation. Therefore, there is no evaluation bias caused by "blurry or misaligned masks".

(2) The edge detail labels and body labels mentioned in the region decoupling strategy are generated entirely based on clear algorithmic procedures and are thus not affected by the subjective judgments of annotators or the output characteristics of different diffusion models.
1) **Body map generation**: First, Gaussian blur is applied to the mask to smooth edges, then the distance transform algorithm is used to calculate the Euclidean distance from each pixel in the mask region to the nearest background edge, and finally, the obtained distance map is normalized and square-rooted, with distance values linearly mapped to the range of [0-255].
2) **Edge detail map generation**: Subtract the body region from the complete tampering mask. The generated body map has the highest brightness in the central part of the tampered region, with brightness gradually decreasing from the center to the edges, reflecting the core part inside the tampered region away from the boundary. The highlighted parts of the detail map are concentrated near the edge contours of the tampered region.

4. Issues Related to Deleted Redundant Modules

R1W3 The Multi-Level Contrastive Learning resembles existing pixel-level contrastive schemes from CFL-Net (WACV 2023) and MPC[1], which makes me believe that there's limited contribution, from my point of view.

R1W7 At line 199, the paper mentions a semi-hard example sampling strategy (SHES) and even shows it in Figure 4, but no textual explanation or citation is provided.

R1W8 At line 202, a fixed configuration of 512 negative samples + 10 positive samples per class is adopted for constructing the memory bank. However, no motivation or sensitivity analysis is offered. Considering that forgery regions vary greatly in size, a fixed sampling ratio may bias the representation. Without justification, these design choices appear arbitrary and may limit reproducibility.

R1W6 Figure 2 references modules such as the Region Enhancement Block (REB) that are never described in the main text. Moreover, the paper states that BAB is structurally identical to DAB, but then uses different names without explaining their distinction or specific functionality.

**Response**:
We appreciate the reviewers for their comments.

(1) R1W3, R1W7, and R1W8 all raise questions about the multi-level contrastive learning module. Considering the model's performance and computational complexity comprehensively, we have decided to remove the contrastive learning module to highlight core innovations. Meanwhile, we add a multi-view adaptive weight fusion mechanism, which connects the multi-view feature extraction module and the subject-detail decoupled localization module to enhance the core modules of the paper.

(2) R1W6: REB has the same structure as the Edge Enhancement Block (EEB), and BAB is identical to DAB in structure, which was introduced in Section III.B of the original paper. We have removed REB and EEB to streamline the model. BAB and DAB act on the body and detail branches respectively, and both modules use a multi-scale feature capture mechanism to enhance the perception ability of the body and detail parts. Since they have the same structure.

---

### Author Response · Authors · 2025-12-04
**Summary of Questions Answered to AC**

5. Issues Related to Dataset Description

R1W13 The DMI dataset creation process is not well-documented. The specific prompts, diffusion model versions (e.g., Stable Diffusion 1.5 vs SDXL), and hyperparameters are not stated and not included in the supplementary, thus I believe the dataset description lacks transparency and reproducibility.

R2W2 Limited Discussion on Adversarial Robustness The paper emphasizes robustness to distribution shifts but does not test whether DMIL-Net is vulnerable to adversarial attacks (e.g., small perturbations designed to fool forgery detectors). Given the adversarial nature of diffusion-based forgeries (e.g., attackers may optimize for stealthiness), this is a critical gap. Improvement: Evaluate the model under adversarial conditions (e.g., FGSM or PGD attacks) and propose mitigation strategies (e.g., adversarial training or input purification).

R4W3 The DMI Dataset, one of the article's innovations, doesn't quite fit the description in the Experiment section. Secondly, DMI simply aggregates existing image editing models to construct it, making it seem like an engineering exercise rather than an innovation.

**Response**:
We have added more detailed supplementary explanations of the dataset creation details to highlight the innovations of the paper in dataset construction.

(1) We appreciate the reviewers for raising this issue and have provided more detailed supplementary explanations of the dataset creation process. It should be noted that datasets generated by the PP method have low quality with obvious visible artifacts. Meanwhile, we found that removing PP from the DMI dataset during training basically does not affect the model performance. Therefore, it was removed during training. The current dataset includes 4 diffusion model generation methods, with hyperparameters using the default values in the paper.
(1) BrushNet SD1.5 version: Steps: 50, guidance scale: 7.5, BrushNet Scale: 1.0 (2) Paint by Example SD v1.4: Steps: 50, guidance scale: 5 (3) Inpaint Anything SD1.5: Steps: 50, guidance scale: 7.5 (4) Repaint: Steps: 250, Resampling times: 10, Jump size: 10

(2) In the DMI dataset, we only retained tampered images and ground-truth masks, not prompt texts. However, we created a high-quality dataset of 4,000 images for diffusion model feature analysis using 4 diffusion models, retaining prompt texts, real images, tampered images, and ground-truth masks. We will make the DMI dataset and the high-quality dataset publicly available for researchers to use.

(3) Datasets are the foundation for studying the features of diffusion-model-forged images, and constructing high-quality datasets is inherently challenging. The innovation of DMI lies in its systematicness, richness, and scalability. We integrated four mainstream diffusion models with different algorithm principles and implementations, covering five popular tampering types (e.g., example-guided, text-guided), and finally generated 40,000 high-quality tampered images. To increase detection difficulty and simulate real scenarios, we introduced a large number of post-processing operations, which enhanced the challenge and practicality of the dataset.

6. Issues Related to Supplementary Computational Efficiency and Evaluation Metrics

R1W10 There is no analysis of computational efficiency, parameter count, or inference time, making the method difficult to reproduce or compare fairly.

R3Q2 Computational Cost of Multi-View Features The multi-view feature learning (RGB, noise, high-frequency domains) improves accuracy but adds preprocessing steps (e.g., Laplacian filtering). What is the trade-off between accuracy gains and inference speed/memory usage compared to simpler baselines (e.g., using only RGB features)?

R2W3 The experimental evaluation needs to be more comprehensive. Only the F1 score is reported, while key metrics such as AUC and IoU, commonly used in IFDL tasks, are missing. This makes the evaluation less comprehensive.

**Response**:
We appreciate the reviewers' suggestions. We have added analyses of computational efficiency, parameter count, and inference time to the comparative experiments and ablation experiments to enhance the completeness and fairness of the experiments. Meanwhile, we have added key metrics such as AUC and IoU. Due to space constraints of the paper, we use F1 and AUC metrics.

7. Issues Related to Supplementary Comparative Experiments

R1W11 The paper presents an incomplete comparison with recent state-of-the-art methods. The comparison excludes several relevant and competitive models that directly address AI-based or diffusion forgeries, such as CAT-Net, TruFor (CVPR 2023), IML-ViT (AAAI 2024), Mesorch [2] (AAAI 2025), and SparseViT[3](AAAI 2025). Without these, it isn't easy to assess the real advancement of DMIL-Net relative to the current state of the art. The claim of superiority is therefore not fully convincing.

---

### Author Response · Authors · 2025-12-04
**Summary of Questions Answered to AC**

R3W4 Incomplete Comparison to Non-Diffusion-Based Forgery Detection Methods While the paper compares DMIL-Net to state-of-the-art diffusion-specific methods, it does not benchmark against traditional forgery detection approaches (e.g., image splicing detection or GAN-based forgery localization). This omission is significant because some traditional methods (e.g., detecting noise inconsistencies or illumination artifacts) might still perform reasonably well on diffusion forgeries, especially for simpler tampering types like object removal.

**Response**: We appreciate the reviewers' suggestions

(1) We have conducted supplementary experiments with the latest methods to demonstrate the superiority of our algorithm.

(2) Regarding the second question, our comparative methods include traditional forgery detection approaches: MVSS and TANet are traditional forgery detection methods for copy-move, splicing, and inpainting; EMF-Net is a method specifically designed for text image tampering; MFI-Net is a deep learning-based general image tampering localization network. Among them, MVSS and EMF-Net utilize noise inconsistency and boundary artifact features.
## BN 数据集

| Methods | F1 | AUC | IoU | Total Parameters | FLOPs | Inference Time (ms) | FLOPs savings | performance loss |
|---------|----|----|----|------------------|-------|---------------------|-----------|----------|
| TA-Net | 0.8743 | 0.9925 | 0.7991 | 62.21M | 51.11G | 16.68 | 0.00%↑ | 0.00% |
| MFI-Net | 0.8670 | 0.9885 | 0.7813 | 59.58M | 12.87G | 24.31 | 74.82% | 0.83% |
| CFL-Net | 0.7930 | 0.9771 | 0.6935 | 167.12M | 78.29G | 13.76 | 53.17%↑ | 9.30% |
| EMF-Net | 0.9015 | 0.9950 | 0.8343 | 96.64M | 34.47G | 20.78 | 32.55% | 3.11%↑ |
| MVSS-Net | 0.7835 | 0.9818 | 0.6816 | 146.81M | 77.31G | 14.77 | 51.27%↑ | 10.39% |
| Mesorch | 0.6943 | 0.9651 | 0.5923 | 85.75M | 145.93G | 23.11 | 185.58%↑ | 20.59% |
| SparseViT | 0.8603 | 0.9938 | 0.7720 | 50.35M | 41.46G | 31.89 | 18.89% | 1.60% |
| CAT | 0.8371 | 0.9989 | 0.7280 | 114.26M | 134.10G | 64.41 | 162.37%↑ | 4.25% |
| DMIL-Net | **0.9140** | **0.9969** | **0.8508** | 79.84M | 41.94G | 32.49 | 17.94% | 4.54%↑ |

## PP 数据集

| Methods | F1 | AUC | IoU | Total Parameters | FLOPs | Inference Time (ms) | FLOPs savings | performance loss |
|---------|----|----|----|------------------|-------|---------------------|-----------|----------|
| TA-Net | 0.7819 | 0.9726 | 0.7162 | 62.21M | 51.11G | 21.16 | 0.00%↑ | 0.00% |
| MFI-Net | 0.8754 | 0.9837 | 0.8270 | 59.58M | 12.87G | 21.44 | 74.82% | 11.96%↑ |
| CFL-Net | 0.7605 | 0.9510 | 0.6969 | 167.12M | 78.29G | 14.86 | 53.17%↑ | 2.74% |
| EMF-Net | 0.8799 | 0.9845 | 0.8363 | 96.64M | 34.47G | 17.86 | 32.55% | 12.53%↑ |
| MVSS-Net | 0.7719 | 0.9557 | 0.7059 | 146.81M | 77.31G | 15.81 | 51.27%↑ | 1.28% |
| Mesorch | 0.6898 | 0.9506 | 0.6244 | 85.75M | 145.93G | 22.51 | 185.58%↑ | 11.78% |
| SparseViT | 0.9339 | 0.9971 | 0.8951 | 50.35M | 41.46G | 30.72 | 18.89% | 19.44%↑ |
| CAT | 0.8858 | 0.9989 | 0.8032 | 114.26M | 134.10G | 49.54 | 162.37%↑ | 13.29%↑ |
| DMIL-Net | 0.9333 | 0.9935 | 0.8996 | 79.84M | 41.94G | 32.49 | 17.94% | 19.36%↑ |

## EI 数据集

| Methods | F1 | AUC | IoU | Total Parameters | FLOPs | Inference Time (ms) | FLOPs savings | performance loss |
|---------|----|----|----|------------------|-------|---------------------|-----------|----------|
| TA-Net | 0.8263 | 0.9730 | 0.7802 | 62.21M | 51.11G | 15.78 | 0.00%↑ | 0.00% |
| MFI-Net | 0.8466 | 0.9679 | 0.8040 | 59.58M | 12.87G | 25.04 | 74.82% | 2.46%↑ |
| CFL-Net | 0.6673 | 0.9158 | 0.6164 | 167.12M | 78.29G | 14.24 | 53.17%↑ | 19.24% |
| EMF-Net | 0.8878 | 0.9865 | 0.8526 | 96.64M | 34.47G | 15.22 | 32.55% | 7.44%↑ |
| MVSS-Net | 0.7277 | 0.9467 | 0.6783 | 146.81M | 77.31G | 15.01 | 51.27%↑ | 11.93% |
| Mesorch | 0.6471 | 0.9450 | 0.5907 | 85.75M | 145.93G | 22.59 | 185.58%↑ | 21.69% |
| SparseViT | 0.9378 | 0.9969 | 0.9016 | 50.35M | 41.46G | 31.50 | 18.89% | 13.49%↑ |
| CAT | 0.8903 | 0.9999 | 0.8073 | 114.26M | 134.10G | 52.93 | 162.37%↑ | 7.75%↑ |
| DMIL-Net | 0.9142 | 0.9922 | 0.8804 | 79.84M | 41.94G | 32.49 | 17.94% | 10.64%↑ |

---

### Author Response · Authors · 2025-12-04
**Summary of Questions Answered to AC**

8. Issues Related to Supplementary Ablation Experiments

R1W12 In experiments, Table 1 and Table 2 show extremely small gains between configurations. Such marginal improvements do not convincingly demonstrate the necessity or effectiveness of each proposed block.

R4W4 In ablation experiments, the improvement in localization performance across various modules is unclear. For example, the effect of NRE can only be seen by comparing Scheme 2 and Scheme 3, but it's unclear whether this improvement is related to the presence of HFE. It would be better to add NRE directly to Scheme 1 and conduct a comparison. Similar experimental verification may also be required for other modules.

**Response**: We appreciate the reviewers' suggestions. We have removed redundant modules and enhanced the performance of core modules. We have changed the design strategy of ablation experiments and added analysis of variance to the ablation experiments. The experimental results demonstrate the effectiveness of the modules.
| Methods     | Total Parameters | FLOPs  | Inference Time (ms) | BN（F1±std/IoU±std/AUC±std）| IA（F1±std/IoU±std/AUC±std）| EI（F1±std/IoU±std/AUC±std）| PP（F1±std/IoU±std/AUC±std）| FLOPs savings |performance loss |
|--------------|----------|--------|--------------|---------------------------------------|---------------------------------------|---------------------------------------|---------------------------------------|-------------------------|----------------------------------|
| nobd         | 74.60M   | 29.18G | 23.08ms      | 0.8554±0.0028/0.7649±0.0034/0.9954±0.0003 | 0.7921±0.0030/0.6887±0.0047/0.9891±0.0007 | 0.9071±0.0033/0.8533±0.0038/0.9973±0.0005 | 0.8818±0.0046/0.8295±0.0053/0.9947±0.0006 | +12.76G                 | 0.0662                           |
| nofreq       | 54.13M   | 32.30G | 30.25ms      | 0.9120±0.0019/0.8467±0.0024/0.9977±0.0002 | 0.8997±0.0032/0.8376±0.0038/0.9946±0.0007 | 0.9096±0.0045/0.8715±0.0048/0.9943±0.0008 | 0.9430±0.0033/0.9135±0.0037/0.9973±0.0006 | +9.64G                  | 0.0079                           |
| nomuifu      | 87.63M   | 45.84G | 28.36ms      | 0.9135±0.0021/0.8503±0.0024/0.9968±0.0004 | 0.8999±0.0029/0.8353±0.0039/0.9934±0.0006 | 0.9131±0.0045/0.8787±0.0046/0.9944±0.0007 | 0.9295±0.0039/0.8958±0.0040/0.9947±0.0006 | -3.90G                  | 0.0092                           |
| nonoise      | 54.13M   | 32.31G | 25.27ms      | 0.9064±0.0023/0.8398±0.0027/0.9969±0.0005 | 0.8864±0.0036/0.8207±0.0040/0.9934±0.0008 | 0.8850±0.0054/0.8451±0.0053/0.9904±0.0010 | 0.8946±0.0051/0.8565±0.0055/0.9850±0.0016 | +9.63G                  | 0.0271                           |
| nonoisefreq  | 27.63M   | 22.67G | 12.23ms      | 0.9082±0.0021/0.8418±0.0026/0.9969±0.0004 | 0.8876±0.0034/0.8215±0.0038/0.9930±0.0008 | 0.8811±0.0051/0.8460±0.0052/0.9909±0.0010 | 0.8937±0.0051/0.8546±0.0055/0.9866±0.0013 | +19.27G                 | 0.0275                           |
| DMIL         | 79.84M   | 41.94G | 32.49ms      | 0.9140±0.0020/0.8508±0.0024/0.9969±0.0004 | 0.9019±0.0030/0.8388±0.0039/0.9937±0.0006 | 0.9142±0.0045/0.8804±0.0047/0.9922±0.0010 | 0.9333±0.0036/0.8936±0.0038/0.9935±0.0009 | -                       | -                                |


9. Issues Related to Supplementary Scalability Experiments

R3W1 Limited Evaluation on Real-World Data and Generalization to Unseen Manipulation Types While the DMI dataset is comprehensive for diffusion-based forgeries, it is entirely synthetic. The experiments focus exclusively on this synthetic data, leaving open critical questions about the method’s performance on real-world tampered images (e.g., from social media or forensic casework). Additionally, the synthetic forgeries in DMI are generated via only five diffusion methods and three tampering patterns (object removal, text-guided editing, shape-aware manipulation). Real-world scenarios may involve hybrid manipulations (e.g., combining object removal with style transfer) or novel diffusion variants not included in the training set. The paper does not address how DMIL-Net might generalize to such cases, nor does it provide results on existing real-world forensic datasets (e.g., Columbia or MICC datasets adapted for diffusion forgeries).

---

### Author Response · Authors · 2025-12-04
**Summary of Questions Answered to AC**

**Response**: According to the reviewers' suggestions, we evaluated the model trained on DMI on the Auto dataset. This dataset uses the DALL-E2 model to automatically generate and stitch mask regions based on text prompts to achieve local or global image forgery. DALL-E2 meets the reviewers' requirement of an unseen diffusion model. Meanwhile, forging content through stitching aligns with the hybrid tampering in real-world scenarios as required by the reviewers. The experimental results demonstrate the generalizability of the model.
The IMD dataset we used in the scalability experiment is a real dataset. The Auto dataset uses the DALL-E2 model to automatically generate and concatenate masked regions based on text prompts, achieving local or global image forgery. DALL-E2 meets the reviewer's requirements for an unseen diffusion model. The experimental results in Table 4 demonstrate that the model has scalability


10. Issues Related to Robustness Experiments

R3W6 Limited Discussion on Adversarial Robustness The paper emphasizes robustness to distribution shifts but does not test whether DMIL-Net is vulnerable to adversarial attacks (e.g., small perturbations designed to fool forgery detectors). Given the adversarial nature of diffusion-based forgeries (e.g., attackers may optimize for stealthiness), this is a critical gap. Improvement: Evaluate the model under adversarial conditions (e.g., FGSM or PGD attacks) and propose mitigation strategies (e.g., adversarial training or input purification).

**Response**:Thank you for the reviewer's suggestions. We have systematically evaluated the robustness of the model to common image disturbances such as Gaussian noise, Gaussian filtering, and scale transformation in section 4.7 of the paper. The experimental results show that DMIL Net can maintain leading and stable localization performance under these traditional image distortion interferences, which preliminarily proves that our model has certain practical robustness. We fully agree that evaluating the robustness of a model to adversarial attacks is crucial. We conducted adversarial attack experiments and preliminary analysis showed that our localization model exhibits potential vulnerability to attacks such as FGSM/PGD on diffusion generated images. Our next step will delve into the fundamental mechanism of its failure, and we will use this analysis as a guide to design targeted defense strategies to enhance the robustness of the model.

11. Format Issues

R1W9 At line 289, there is an invalid or unresolved reference [28], which should be corrected. Besides, several formulae are overloaded and lack intermediate explanations, making it difficult to follow the data flow.

**Response**: We appreciate your suggestions. We have corrected the invalid reference and conducted a full-text check. We will restructure and format the paper to highlight and clarify core formulas, remove redundant content, and emphasize the core innovations of the paper.

### B. Summary of Questions

1. Questions

R2Q1: How does DMIL-Net perform at the image level? Does it tend to produce a high false positive rate?

This paper specifically addresses the pixel-level localization problem of locally forged images based on diffusion models. Therefore, the model does not directly provide image-level binary classification judgments. For future work, we will leverage multi-modal large models and combine the low-level forensic features required for tampering detection that large models lack to construct a unified model framework capable of achieving image-level and pixel-level detection, as well as generating text descriptions of the model's discrimination process based on logical reasoning.

R4Q1: Line 174，should be corrected.

We appreciate you pointing out this issue, and we have made the correction.

---

### Author Response · Authors · 2025-12-04
**Summary of Questions Answered to AC**

2. Questions Already Answered in Section A (Weaknesses)

R1Q1 Why are some submodules like REB not described, while BAB and DAB are said to be identical? What is their functional distinction?
Please refer to A4

R1Q2 Could you explain the rationale for using a fixed 512/10 sampling in the memory bank? Did you test variable ratios depending on the forgery size?
Please refer to A4

R1Q3 What exactly is the “semi-hard example sampling strategy” (SHES) in Figure 4? Please describe it explicitly.
Please refer to A4

R1Q4 Please correct the missing citation [28] at line 289 and verify all other references.
Please refer to A11

R1Q5 How do the small performance increments in Tables 1 and 2 support the claimed effectiveness of each module? I'm not sure such a heavy design is helpful. Can you report confidence intervals or variance?
Please refer to A8

R1Q6 Why are key SOTA methods (CAT-Net, TruFor, IML-ViT, SparseViT) omitted from comparison?
Please refer to A7

R1Q7 Will the DMI dataset, including prompt texts and model versions, be released for public use?
Please refer to A5

P3Q1 Real-World Generalization and Unseen Tampering Types The experiments focus entirely on synthetic data from five diffusion methods and three tampering patterns. How does DMIL-Net perform on real-world tampered images (e.g., social media or forensic datasets) or hybrid/unknown manipulation types (e.g., diffusion-based forgeries combined with traditional splicing)?
Please refer to A9

P3Q2 Computational Cost of Multi-View Features The multi-view feature learning (RGB, noise, high-frequency domains) improves accuracy but adds preprocessing steps (e.g., Laplacian filtering). What is the trade-off between accuracy gains and inference speed/memory usage compared to simpler baselines (e.g., using only RGB features)?
Please refer to A6

P3Q3 Theoretical Justification for Region Decoupling The region decoupling strategy (body-edge separation) is empirically effective, but the paper lacks a theoretical explanation for why this approach improves localization. Is there a principled connection between diffusion model properties and the decoupling mechanism?
Please refer to A2

P3Q4 Comparison to Traditional Forgery Detection Methods The paper compares DMIL-Net to diffusion-specific methods but omits traditional forgery detection techniques (e.g., noise inconsistency or illumination-based methods). Are these methods competitive on diffusion forgeries, and if not, why?
Please refer to A7

P3Q5 Annotation Consistency in the DMI Dataset The DMI dataset defines "tampered edges" but does not clarify how annotations were created (e.g., manual vs. automated). How consistent are the edge annotations across annotators or diffusion methods?
Please refer to A3

P3Q6 Adversarial Robustness of DMIL-Net The paper claims robustness to distribution shifts but does not test adversarial attacks (e.g., FGSM or PGD). Could diffusion-based forgeries be optimized to evade DMIL-Net?
Please refer to A10

P3Q7 Interpretability of Multi-Level Contrastive Learning The multi-level contrastive learning module models cross-modal dependencies but does not explain how these dependencies correlate with tampering artifacts. Can the authors provide visualizations or feature maps to interpret what the module learns?
Please refer to A4

---

### Meta-Review · Area_Chair_kPmD · 2026-01-02

**Summary:**

The reviewers have raised substantial concerns about this submission. The primary issues center around limited methodological novelty, overly complex architecture without clear motivation, insufficient theoretical justification for the proposed region decoupling strategy, and questions about real-world applicability. Reviewer 7GmT provided the most critical assessment, noting that the framework integrates existing components (multi-view fusion, contrastive learning, multi-branch decoders) without introducing distinctive new insights. The architecture was described as "overly complicated and poorly motivated" with numerous unexplained modules. Additional concerns included incomplete comparisons with state-of-the-art methods, marginal improvements in ablation studies, insufficient dataset documentation for reproducibility, and lack of computational efficiency analysis. Reviewers qXSa and 8n7g, while more positive about the originality and dataset contribution, still highlighted significant weaknesses regarding real-world generalization, theoretical foundation, and methodological clarity.

**Reviewer Concerns:**

The authors have made reasonable attempts to address some technical concerns in their rebuttal. They have removed redundant modules, added computational efficiency analysis and additional evaluation metrics, expanded comparisons to include more SOTA methods, provided more detailed dataset documentation, and conducted preliminary adversarial attack experiments. They have also added variance analysis to ablation studies and clarified the dataset construction process.

However, several fundamental concerns remain outstanding and are not adequately addressed:
1. The core criticism that DMIL-Net primarily integrates existing techniques without substantial conceptual innovation remains valid. While the authors argue their approach is tailored to diffusion model artifacts, reviewers correctly note that multi-view fusion and hierarchical decoders are well-established in image forensics.
2. The region decoupling strategy lacks a principled theoretical foundation. The authors' explanation remains empirical rather than theoretical, failing to establish why this approach is particularly suited to diffusion-generated forgeries beyond heuristic arguments.
3. The evaluation still focuses predominantly on synthetic data. While the authors added some cross-dataset testing, concerns about performance on real-world tampered images from social media or forensic casework remain unaddressed.

**Reviewer Scores:**

After rebuttal, all reviewers maintained their original scores. The paper represents a solid engineering effort with thorough experimentation, but it falls short of the novelty and theoretical contributions expected for ICLR. The work would benefit from more rigorous theoretical analysis, clearer differentiation from existing methods, and stronger demonstration of real-world applicability before being suitable for publication at a top-tier conference.

---

### Decision · Program_Chairs · 2026-01-26

Reject